# PIP2 depletion and altered endocytosis caused by expression of Alzheimer's disease-protective variant PLCγ2 R522

Emily Maguire[1,†], Georgina E Menzies[1,2,†] (ID), Thomas Phillips[1,†], Michael Sasner[3], Harriet M Williams[3], Magdalena A Czubala[4], Neil Evans[1], Emma L Cope[2], Rebecca Sims[5], Gareth R Howell[3], Emyr Lloyd-Evans[2], Julie Williams[1,5,‡], Nicholas D Allen[2,‡] (ID) & Philip R Taylor[1,4,*,‡] (ID)

## Abstract

Variants identified in genome-wide association studies have implicated immune pathways in the development of Alzheimer's disease (AD). Here, we investigated the mechanistic basis for protection from AD associated with PLCγ2 R522, a rare coding variant of the *PLCG2* gene. We studied the variant's role in macrophages and microglia of newly generated *PLCG2*-R522-expressing human induced pluripotent cell lines (hiPSC) and knockin mice, which exhibit normal endogenous *PLCG2* expression. In all models, cells expressing the R522 mutation show a consistent non-redundant hyperfunctionality in the context of normal expression of other PLC isoforms. This manifests as enhanced release of cellular calcium ion stores in response to physiologically relevant stimuli like Fc-receptor ligation or exposure to Aβ oligomers. Expression of the PLCγ2-R522 variant resulted in increased stimulus-dependent $PIP_2$ depletion and reduced basal $PIP_2$ levels *in vivo*. Furthermore, it was associated with impaired phagocytosis and enhanced endocytosis. PLCγ2 acts downstream of other AD-related factors, such as TREM2 and CSF1R, and alterations in its activity directly impact cell function. The inherent druggability of enzymes such as PLCγ2 raises the prospect of PLCγ2 manipulation as a future therapeutic approach in AD.

**Keywords** Alzheimer's disease; microglia; phagocytosis; PIP2; PLCG2
**Subject Categories** Molecular Biology of Disease; Neuroscience
**The EMBO Journal (2021) 40: e105603**

## Introduction

The gene encoding phospholipase C gamma 2 (PLCγ2) was recently linked by whole-exome microarray genome-wide association (rs72824905/P522R, $P = 5.38 \times 10^{-10}$) to Alzheimer's disease (AD), the most common form of dementia (Qiu *et al*, 2009). The majority of AD cases are late-onset AD (LOAD) with a sporadic aetiology. The heritability of AD is estimated at 58–79% (Gatz *et al*, 2006), demonstrating a strong genetic influence on the development of AD. Genome-wide association studies (GWAS) have identified genomic regions associated with LOAD (Harold *et al*, 2009; Hollingworth *et al*, 2011; Lambert *et al*, 2013; Sims *et al*, 2017; Sims *et al*, 2020). As well as these common variants, a number of rare genetic variants have been reported which implicate immunity and a crucial role for microglia in disease development. The rare protective R522 protein-coding variant in *PLCG2* is one such example (Sims *et al*, 2017). PLCγ2 is the first classically drug-targetable molecule identified from a LOAD genetic study. Understanding how this variant may lower the risk of developing AD is essential, not just for a greater understanding of disease mechanisms, but also for guiding future drug development approaches. Fundamental questions remain unresolved: How might the variant alter PLCγ2 function? Does the R522 variant manifest a functional alteration in physiologically relevant cells (such as microglia)? When expressed at endogenous expression levels does this alteration occur at the level of cell signalling? Is its function non-redundant in the context of expression of other additional PLC isoforms as found in microglia? This study sought to address these questions.

PLCγ2 is a member of the phospholipase C (PLC) family of enzymes. All 13 members of the PLC enzyme family play a role in signal transduction pathways (Kadamur & Ross, 2013), and there are two PLC gamma molecules, 1 and 2, which share a high

1  UK Dementia Research Institute at Cardiff, Cardiff, UK
2  School of Biosciences, Cardiff University, Cardiff, UK
3  The Jackson Laboratory, Bar Harbor, ME, USA
4  Systems Immunity University Research Institute, Cardiff, UK
5  MRC Centre for Neuropsychiatric Genetics & Genomics, Cardiff, UK
   *Corresponding author (lead contact). Tel: +44 2920687328; E-mail: taylorpr@cardiff.ac.uk.
   †These authors contributed equally to this work as first/second/third authors
   ‡These authors contributed equally to this work as senior authors

sequence homology but differ greatly in their expression profiles. PLCγ2 is largely considered to be expressed in haematopoietic cells and is involved in the regulation of both development and function of various haematopoietic cells (Koss *et al*, 2014). In the brain, PLCγ2 is expressed predominantly in microglia. PLCγ2 has been shown to catalyse the hydrolysis of phosphatidylinositol(4,5)bisphosphate [PI(4,5)P$_2$] and this serves to increase concentrations of cytosolic facing diacylglycerol (DAG) and inositol 1,4,5 trisphosphate (IP$_3$), which in turn increases the concentration of intracellular Ca$^{2+}$ (Frandsen & Schousboe, 1993; Zheng *et al*, 2009; Kadamur & Ross, 2013). PLCγ enzymes have also been implicated in aberrant cellular responses linked to complex disease development (Everett *et al*, 2009; Bunney & Katan, 2010). One example is an association between dominantly inherited complex immune disorders and gain-of-function mutations in PLCγ2, such as deletions in the PLCG2-associated antibody deficiency and immune dysregulation (PLAID) disorder that occur in the cSH2 autoinhibitory domain (Ombrello *et al*, 2012).

The cSH2 domain of PLCγ2 prevents enzymatic activity, and receptor ligation is believed to displace this domain revealing the active site; it is therefore not surprising that deletions here can confer increased enzymatic activity, in certain conditions (Milner, 2015). Through an ENU mutagenesis strategy, Yu *et al* (2005) identified a gain-of-function variant of PLCγ2 which leads to subsequent hyperactivation of B cells and innate immune cells. They showed an autoimmune and inflammatory response, which implicates PLCγ2 as a key regulator in autoimmune and inflammatory disease. In microglia, PLCγ2 regulates multiple signalling pathways leading to functions such as phagocytosis, secretion of cytokines and chemokines, cell survival and proliferation (Ulrich *et al*, 2014; Jay *et al*, 2015; Wang *et al*, 2015). With this in mind, it is important to consider how PLCγ2 function may effect microglial responses in the context of amyloid deposition and other neuroinflammatory components of dementia seen in AD. Deficiency in TREM2, an upstream receptor that signals through PLCγ2, has been previously linked to increased risk of AD (Guerreiro *et al*, 2013; Sims *et al*, 2020). PLCγ2 also sits downstream of CSF1R, inhibitors or which are currently in trials in the context of Alzheimer's disease. Thus, the role of PLCγ2 in Alzheimer's disease may be complex.

The recently described rare coding R522 variant of PLCγ2 is a protective variant that causes a neutral to positive amino acid change. We hypothesized this mutation causes a structural and functional impact on the protein by altering the behaviour of the loop in which residue 522 is found and subsequently other protein domains. In this manuscript, we use CRISPR gene-edited human iPSC (hiPSC) and mouse models to ask how the protective R522 variant of PLCγ2 influences cell signalling and function in physiologically relevant cells (microglia and macrophages) and at appropriate expression levels does the protective R522 variant of PLCγ2 influence cell signalling and classic functional activities, such as endocytosis? We demonstrate that the R522 variant results in a hyperfunctional enzyme, which in AD relevant cells are associated with enhanced Ca$^{2+}$ signalling, substrate depletion and altered phagocytic and endocytic responses. The relevance of this to protection from AD and the potential consequences for therapeutic approaches are discussed.

## Results

### Molecular dynamic predictions and structural impact of P522R variant

The HoPE server was used to report on the possible effects that the change in amino acid may have on the protein structure, and make some functional predictions (Venselaar *et al*, 2010). There are a number of obvious changes; to begin with, the mutated amino acid (R522) is bigger than the wild type (P522) (Fig 1A), and it also carries a positive charge and is less hydrophobic than the WT. The size change is predicted to cause an interaction between this amino acid and other parts of the protein, and this interaction will cause a knock-on effect on protein structure. HoPE also shows the sPH domain, where the mutation is found, to interact with two other domains which are involved in protein function.

Both P522 and mutated protein simulations have consistent energy, pressure and volume outputs with very low standard errors, showing a good stability in the simulation. The root mean standard deviation (RMSD) can be used to record the overall flexibility or that of individual amino acids. RMSD for the P522 and R522 simulations is not statistically different ($P = 1.83$e-09) indicating the mutation does not have an overall effect on the flexibility of the protein. However, mutation appears to have a greater effect on the flexibility of amino acid 522, along with other small areas of the protein including the majority of the SH2 domains (Fig 1B). A number of these residues show statistically significant differences in flexibility (i.e. $P \leq 0.05$) between the P522 and the R522 proteins.

The R522 mutation has a wider impact on the PLCγ2 protein than just the changing of the local loop structure in which it is found. The SH2 domains are also significantly altered, in both position and structure, where the rest of the structure is conserved (Fig 1). When compared to the wild type, the mutated SH2 domain is shifted far to the left (Fig 1C). Further to this change, hydrogen bonding is increased when proline is switched to an arginine, with an increased number of donors and acceptors. Proline has one acceptor, and arginine has two acceptors and five hydrogen bonding donor sites. During the simulation, P522 formed just two hydrogen bonds with near-by

**Figure 1. Molecular dynamic and structure of P522R variant.**

A Proline structure, arginine structure, and the mutation site image, produced by the HoPE server, depicting the P522 in green with the R522 mutation in red (Venselaar *et al*, 2010).

B Flexibility (RMSD) differences between the wild-type and mutated protein, significant differences are highlighted in the figure. RMSD was measured per residue over the whole simulation time using the GROMACS tool g_rms and here averages were plotted with standard deviations represented in the error bars. Data were analysed using a Mann–Whitney *U*-test in the R statistical package.

C Wild-type and mutated structures overlaid in MOE. In this image, most of the structure is light grey with the SH2 domain highlighted in blue (P522) and orange (R522), and the inset shows a closer view of the SH2 domain with both P522 and R522 variants highlighted.

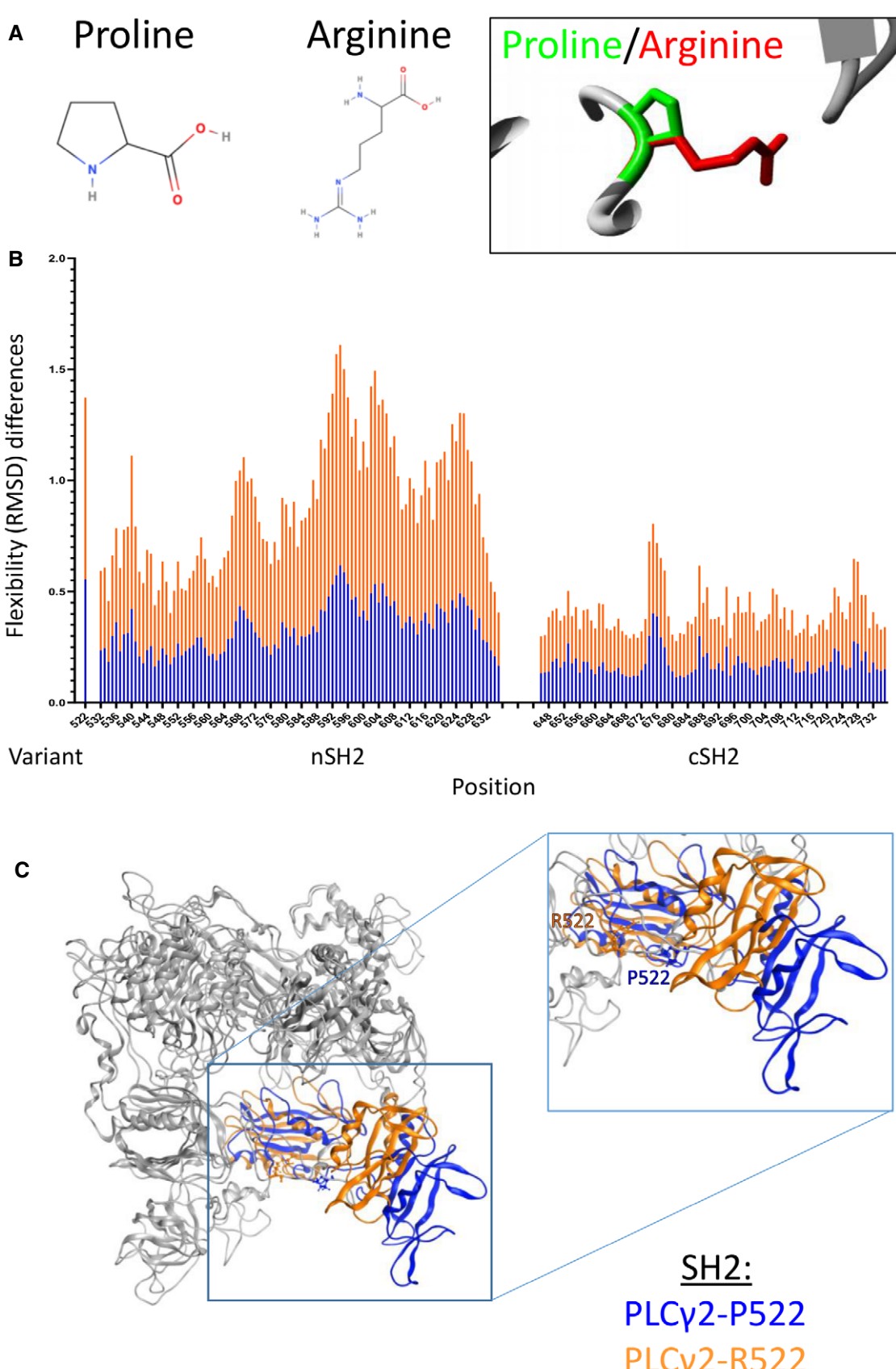

**Figure 1.**

residue, 524. This suggests that the P522 residue has local interactions only. The mutated p.R522 residue, however, formed hydrogen bonds with seventeen different residues (Appendix Tables S1–S3). These changes in interaction cause a previously randomly coiled section of the protein to form a helix (Fig 1). The change in flexibility and position of the SH2 domains could feasibly modulate PLCγ2's ability to trigger intracellular $Ca^{2+}$ release.

### The R522 variant of Plcg2 confers increased receptor-mediated $Ca^{2+}$ release in macrophages and microglia

Four independent conditionally immortalized macrophage precursor (MØP) polyclonal cell lines were derived from both male and female $Plcg2^{P522}$ and $Plcg2^{R522}$ mice. The genotype of donor mice was confirmed prior to use (Appendix Fig S1A and B), and similar expression of $Plcg2$ in M-CSF-differentiated "M-MØP" macrophages of the two genotypes was verified by qPCR (Appendix Fig S1C).

Results from Fura2 intracellular $Ca^{2+}$ assays demonstrated that M-MØPs differentiated from $Plcg2^{R522}$ precursor cell lines produced a significantly greater rise in intracellular $Ca^{2+}$ after exposure to anti-FcγRII/III (Fig 2A, Appendix Fig S1D and E), compared to their Plcg2P522 expressing counterparts. As such, the R522 variant appears to be hyperfunctional compared to WT in this *in vitro* system. All the cell lines produced a similar response after exposure to ionomycin. This suggests both cell lines were alive and capable of $Ca^{2+}$ response.

To confirm the R522 variant is also hyperfunctional in primary cells, microglia were cultured from the cortex and hippocampus of neonatal $Plcg2^{P522}$ and $Plcg2^{R522}$ mice. Using a Fura2 assay, it was shown that anti-FcγRII/III exposure resulted in a greater $Ca^{2+}$ response in the R522-variant expressing cells when compared to the WT. This was found in microglia cultured from both cortex (Fig 2B) and hippocampus (Appendix Fig S1F). The $Ca^{2+}$ response in both regional types was inhibited using edelfosine suggesting the $Ca^{2+}$ response was due to the function of PLCγ.

We next repeated Fura2 intracellular $Ca^{2+}$ assays in microglia-like cells derived from either control PLCγ2R522 or variant PLCγ2R522-expressing hiPSCs to confirm our findings with human cells (Fig 2C). Similar to the mouse cells, activation of PLCγ2 using anti-CD32 (anti-FcγRII) resulted in a greater cytosolic $Ca^{2+}$ increase in PLCγ2R522 variant cells compared to control PLCγ2P522 cells.

We next examined stimulation of PLCγ2 through more physiologically relevant pathways. We examined the $Ca^{2+}$ response of mouse macrophages (Fig 3A) and microglia (Fig 3B) to stimulation with LPS (50 ng/ml) and $A\beta_{1-42}$ oligomers (0.5 μM). Both microglia and macrophages expressing the R522 variant exhibited enhanced $Ca^{2+}$ responses to LPS compared to WT. Whilst the response to 0.5 μM $A\beta_{1-42}$ oligomers was not significantly increased, it was when a 40 μM dose was used (Appendix Fig S1J and K). Using DOPS-Liposomes to activate TREM2, we found increased cytosolic $Ca^{2+}$ release in all three examined models with the R522 variant compared to those with the P522 variant (Fig 3C–E).

In order to confirm that the assays were measuring PLCγ2-induced $Ca^{2+}$ release, a number of approaches were taken, employing both Fura2 and Fluo-8 assays. First, PLCγ was inhibited with edelfosine and U71322 and the IP3 receptor blocked with xestospongin C during stimulation of macrophages with anti-FcγRII/III (2.4G2) (Appendix Fig S1B). All three inhibitors prevented the FcγR-induced $Ca^{2+}$ response (Appendix Fig S1G). To confirm that the intracellular $Ca^{2+}$ release was due to PLCγ2 activity in this assay, GapmeRs were used to knock down $Plcg2$ mRNA in mouse macrophages, prior to study (Appendix Fig S1H). GapmeR-mediated knockdown was very effective (Appendix Fig S1H, left panel) and prevented FcγRII/III-mediated $Ca^{2+}$ release (Appendix Fig S1H, right panel). Similar results were obtained with primary microglia (Appendix Fig S1I). GapmeR knockdown of $Plcg2$ was also effective in reducing calcium responses in macrophages after stimulation with LPS (50 ng/ml) and $A\beta_{1-42}$ oligomers (0.5 μM).

### Expression of the AD protective PLCγ2 R522 variant impaired phagocytic and enhanced endocytic clearance

Phagocytosis can occur via a variety of mechanisms and acts as a key function of microglia and macrophages within the brain (Janda *et al*, 2018). Furthermore, PLCγ2 is downstream of TREM2 and CSF1R, both known to regulate aspects of the cell biology of these lineages, including microglial phagocytosis (Xing *et al*, 2015). We assessed how the R522 variant of PLCγ2 influenced phagocytic activity using the standardized pHrodo-labelled bioparticles derived from *E. coli* and zymosan. Mouse macrophages, microglia and human iPSC-derived microglial (Fig 4A–C, respectively) all exhibited significantly reduced phagocytosis of *E. coli* when expressing the R522 variant compared to the common P522 variant. Similar results were obtained with all three cell types with zymosan particles (Fig 4D–F).

Subsequently, we examined the endocytic clearance of fluorescent $A\beta_{1-42}$ oligomers by the same cells (Fig 5A–C). In all cases, expression of the R522 variant resulted in enhanced clearance of the oligomers. Similar results were obtained with a model cargo (pHrodo-labelled 10 KDa Dextran) (Fig 5D–F).

---

**Figure 2. Receptor-mediated release of $Ca^{2+}$ in macrophages and microglia.**

A  Fura2 340/380 time traces from M-CSF-differentiated macrophages derived from conditionally immortalized macrophage precursor cell lines (M-MØP) of $Plcg2^{P522}$ mice (P522, blue) and $Plcg2^{R522}$ mice (R522, red). One set of cell lines, generated from male mice, is shown. Cells were exposed to 5 μg/ml anti-FcγRII/III along with 20 μM EGTA and 2 μM ionomycin as indicated. Data show the mean ± SD of three independent experiments and were analysed by two-way ANOVA with Sidak post hoc tests (**$P < 0.01$ and ****$P < 0.0001$).

B  Fura2 340/380 time traces from primary microglia derived from the cortex of $Plcg2^{R522}$ mice (blue: P522) and $Plcg2^{R522}$ mice (red: R522) with or without pre-exposure for 2 h with edelfosine (10 μM). Cells were exposed to 5 μg/ml anti-FcγRII/III along with EGTA and 2 μM ionomycin. Data show the mean ± SD of three independent experiments and were analysed by two-way ANOVA with Sidak post hoc tests (**$P < 0.01$; ***$P < 0.001$; and ****$P < 0.0001$).

C  $PLCG2^{R522}$ hiPSC-derived microglia show increased cytosolic $Ca^{2+}$ influx in comparison with controls following activation of PLCγ2 using anti-CD32. Cytoplasmic $Ca^{2+}$ increase following activation of PLCγ2 using anti-CD32. Three independent $PLCG2^{R522}$CRISPR-engineered clones were examined. Data shown are mean ± SD of 4 independent experiments and were analysed by one-way ANOVA with Tukey's multiple comparison test (*$P < 0.05$ and **$P < 0.01$).

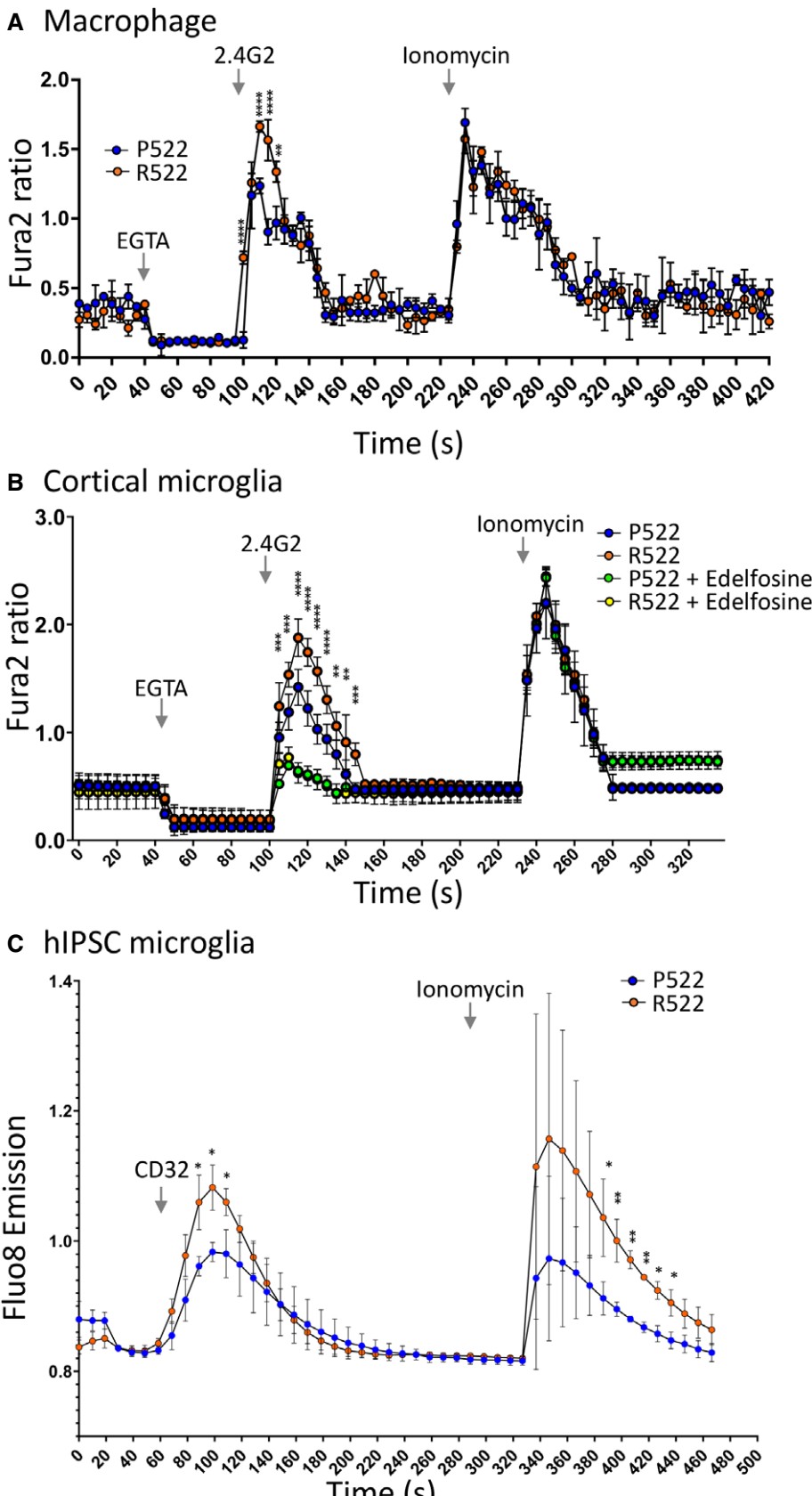

**Figure 2.**

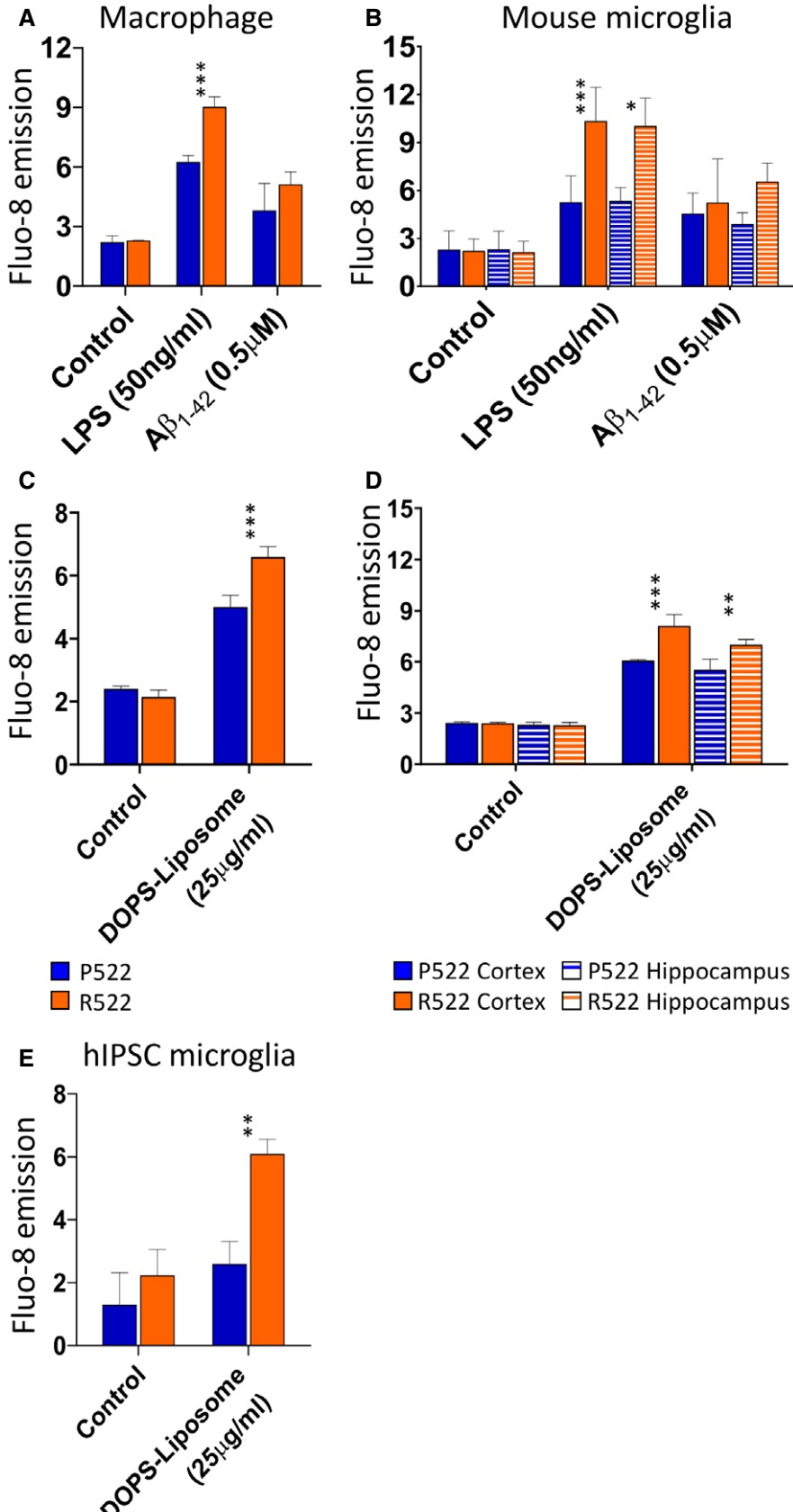

**Figure 3.**

**Figure 3. Response of PLCγ2 in macrophages/microglia to physiologically relevant stimuli.**

A  M-MØP (blue: P522; red: R522) were loaded with Fluo-8 Ca$^{2+}$ indicator and examined for peak changes in fluoresce after exposure to LPS (50 ng/ml) or Aβ$_{1–42}$ oligomers (0.5 μM).

B  Microglia from *Plcg2*$^{P522}$ (blue: P522) mice and *Plcg2*$^{R522}$ mice (red: R522) from neonate cortex (solid colour) or hippocampus (striped) were loaded with Fluo-8 Ca$^{2+}$ indicator. These cells were then examined for peak changes in fluoresce after exposure to LPS (50 ng/ml) or Aβ$_{1–42}$ oligomers (0.5 μM).

C–E  M-MOPS (C), primary mouse microglia (D) and hiPSC (E) (blue: P522 and red: R522) were loaded with Fluo-8 Ca$^{2+}$ indicator and examined for peak changes in fluoresce after exposure to DOPS-liposomes (25 μg/ml).

Data information: Data shown as the mean ± SD of three independent experiments. The data were analysed by two-way ANOVA with Sidak's post-tests (*$P < 0.05$, **$P < 0.01$ and ***$P < 0.001$).

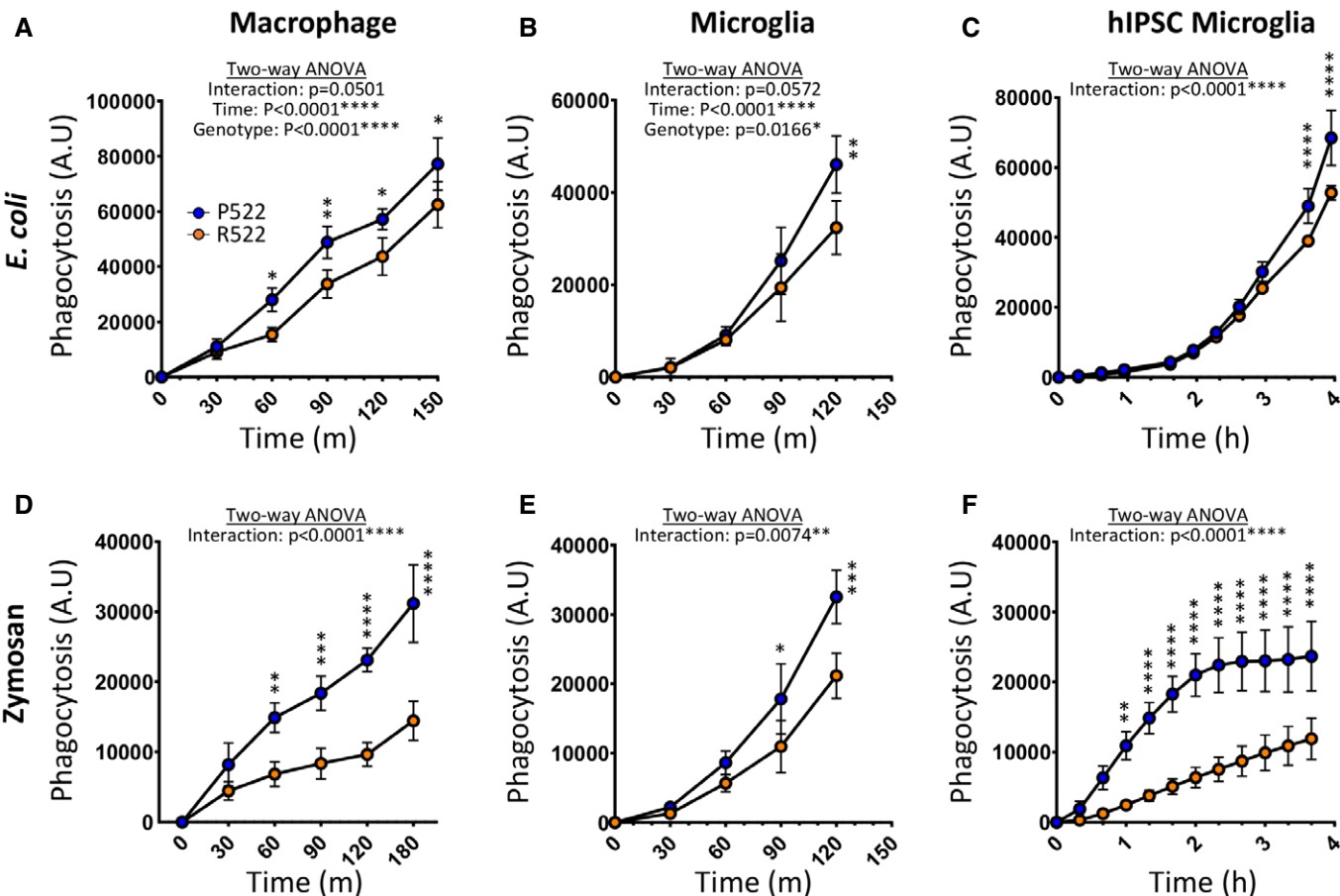

**Figure 4. Protective R522 M-MOP and microglia display decreased phagocytosis of *E. coli* and zymosan when compared with common P522 variant cells.**

A–F  Phagocytotic activity of R522 and P522 M-MOP and microglia was assessed using pHrodo Red *E. coli* and zymosan BioParticles. Phagocytosis arbitrary units (A.U) describe the amount of bioparticle cellular fluorescence emission at each time point. *E. coli* uptake in M-MOP (A), primary mouse microglia (B) and hiPSC-derived microglia (C). Zymosan uptake in M-MOP (D), primary mouse microglia (E) and hiPSC-derived microglia (F). For hiPSC-derived microglia, 3 isogenic P522 and 3 isogenic R522 clones were examined with at least 6 wells in three independent experiments. All microglia and M-MOP data show the mean ± SD of three independent experiments and were analysed by two-way ANOVA using Sidak multiple comparison test. *$P < 0.05$, **$P < 0.01$, ***$P < 0.001$ and ****$P < 0.0001$ (blue: P522 and red: R522).

Inhibition of actin cytoskeleton remodelling using cytochalasin D effectively reduced uptake of all examined cargos by > 90% in mouse macrophages and primary microglia (Appendix Fig S2). Chlorpromazine, an inhibitor of clathrin-mediated endocytosis (Uriarte *et al*, 2009), was used to investigate the role of clathrin-mediated endocytosis in these systems. Chlorpromazine was effective in reducing *E. coli* uptake by ~30% in mouse macrophages and

~20% in microglia (Appendix Fig S2A and E). Zymosan uptake was only slightly inhibited by ~10% in macrophages and not effectively inhibited in primary microglia (Appendix Fig S2B and F). Chlorpromazine inhibited uptake of amyloid oligomers by ~85% in mouse macrophages and microglia (Appendix Fig S2C and G). Similarly, dextran uptake was inhibited by 80% in macrophages and 90% in microglia (Appendix Fig S2D and H). Methyl-β-cyclodextrin, an

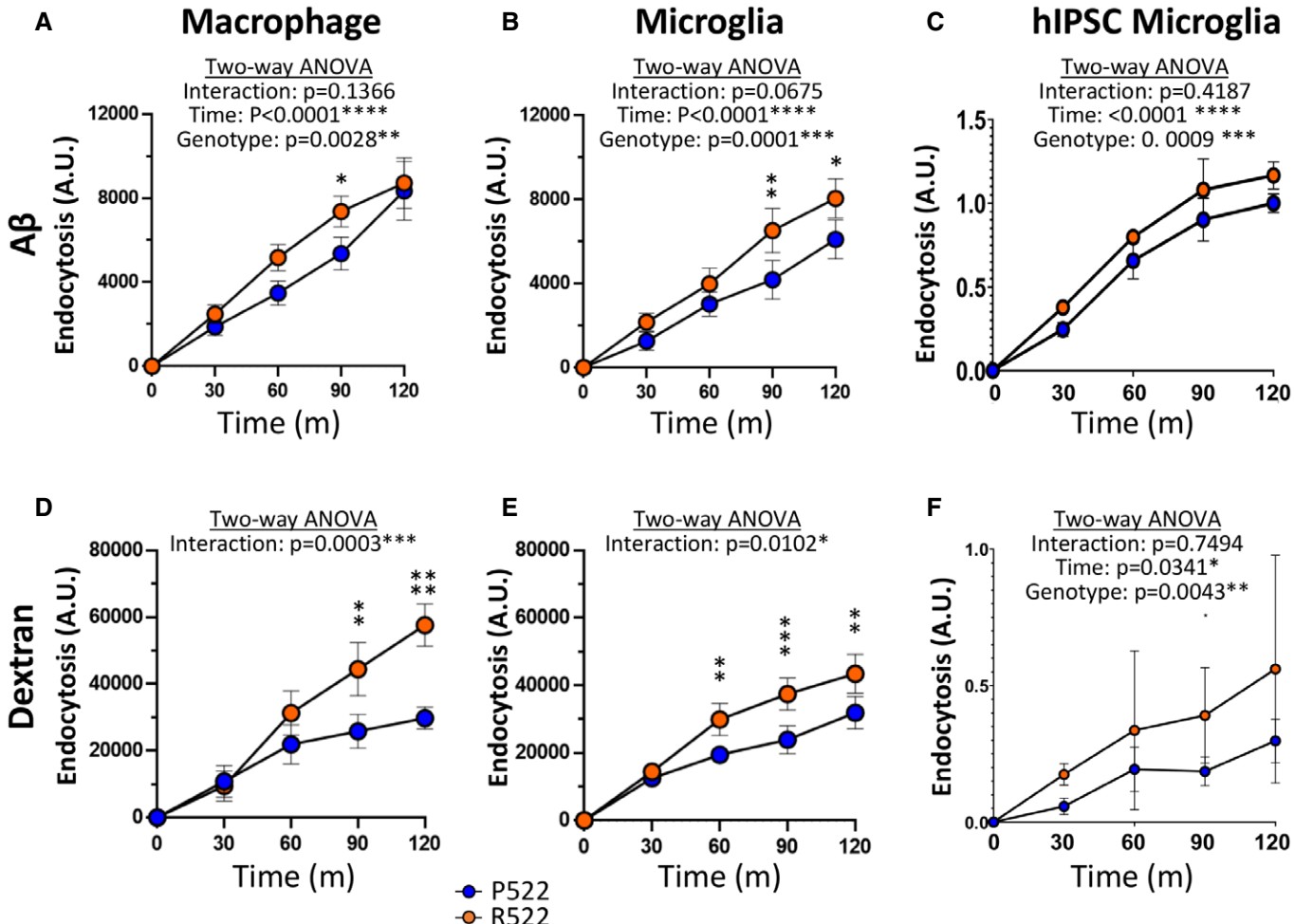

**Figure 5.  Protective R522 M-MOP and microglia display increased endocytosis of soluble Aβ$_{1–42}$ oligomers and dextran when compared with common P522 variant cells.**

A–F  Endocytic activity of R522 and P522 M-MOP and microglia was assessed using FITC soluble Aβ$_{1–42}$ oligomers and pHrodo Red dextran (10,000 MW). Endocytosis arbitrary units (A.U) describe the amount of bioparticle cellular fluorescence emission at each time point. Aβ$_{1–42}$ oligomer uptake in M-MOP (A), primary microglia (B) and hiPSC-derived microglia (C). Dextran uptake in M-MOP (D), primary microglia (E) and hiPSC-derived microglia (F). For hiPSC-derived microglia, 3 isogenic P522 and 3 isogenic R522 clones were examined with at least 6 wells in three independent experiments. All microglia and M-MOP data show the mean ± SD of three independent experiments and were analysed by two-way ANOVA using Sidak's multiple comparison test. *$P < 0.05$, **$P < 0.01$ and ***$P < 0.001$ (blue: P522 and red: R522).

inhibitor of clathrin-independent endocytosis (Sandvig *et al*, 2018), was used to further examine the uptake mechanisms in these systems. Methyl-β-cyclodextrin did not effectively inhibit uptake of zymosan or *E. coli* bioparticles in macrophages or microglia (Appendix Fig S2A, B, E and F). Uptake of oligomers however was decreased by ~50%, and uptake of dextran was decreased by ~70% in macrophages and microglia (Appendix Fig S2C, D, G and H). For all these inhibitors, the effect was similar on cells with both variants.

We next investigated the role of phosphoinositide metabolism on phagocytic and endocytic uptake within our models. GapmeR knockdown of *Plcg2* was able to inhibit the uptake of *E. coli* and zymosan bioparticles in macrophages (Appendix Fig S3A and B). Knockdown of *Plcg2* reduced uptake of amyloid oligomers by ~30% in cells with the P522 variant and ~50% in cells with the R522 variant (Appendix Fig S3C). Similarly, the knockdown reduced uptake

of dextran by ~50% in cells with both variants (Appendix Fig S3D). Increasing PIP$_2$ levels in mouse macrophages using a PIP$_2$ shuttle increased uptake of *E. coli* in cells with both variants but the increase was larger in those cells with the R522 variant (Appendix Fig S3E). Zymosan uptake was only increased by PIP$_2$ supplementation in cells with the R522 variant (Appendix Fig S3F). Increase of PIP$_2$ did not significantly increase uptake of amyloid oligomers or dextran (Appendix Fig S3G and H). Similarly, inhibition of PLCγ2 using various inhibitors and inhibitor of phosphatidylinositol three kinase decreased uptake of *E. coli* particles but not dextran uptake (Appendix Fig S4A and B). Inhibition of PTEN increased uptake of *E. coli* (Appendix Fig S4A and B), whilst SHIP1 inhibition increased uptake of *E. coli* but did not significantly increase dextran uptake. Altering the dosage of the cargo altered the uptake by M-MOP (Appendix Fig S5A–D).

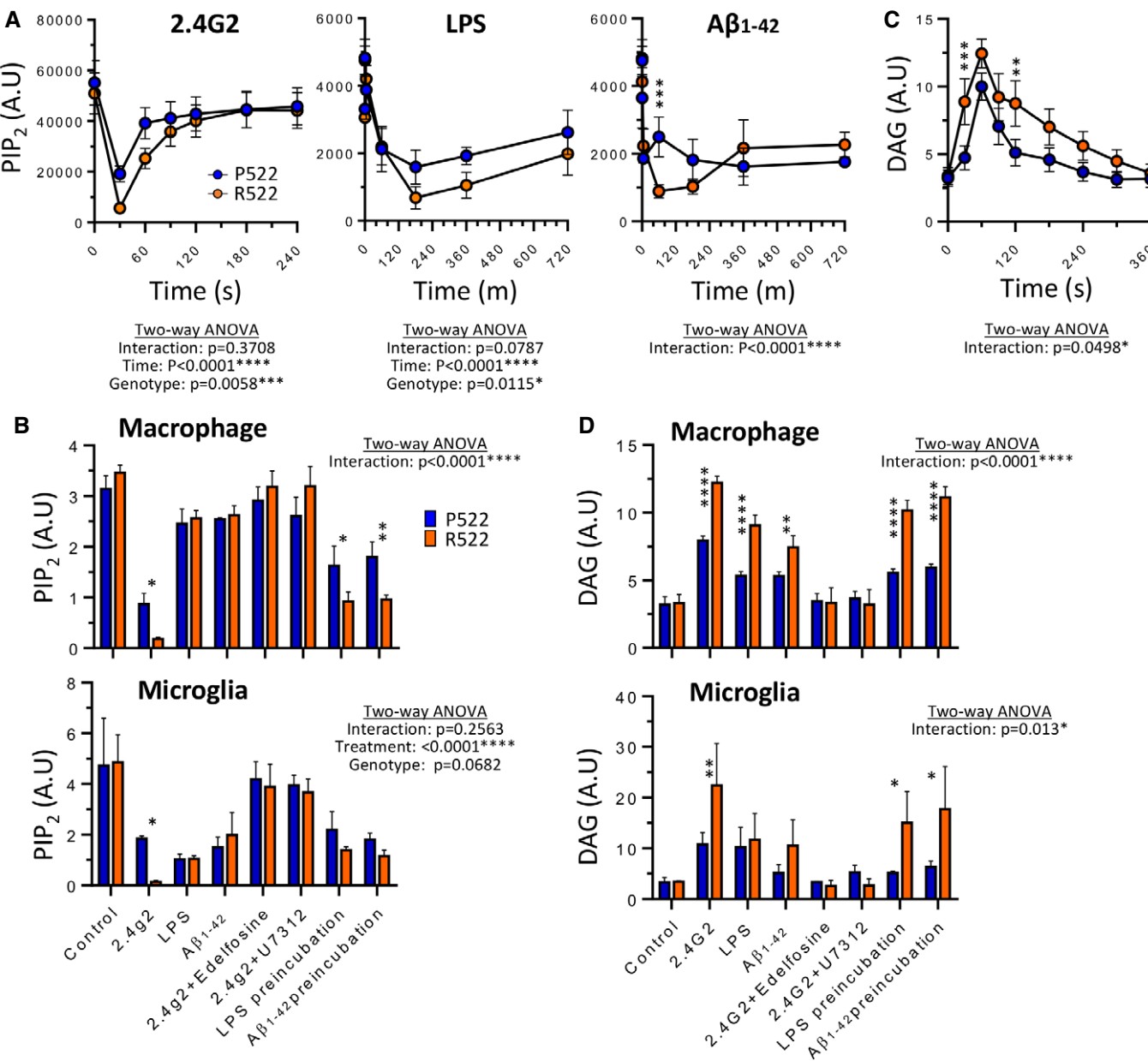

**Figure 6. Protective R522 M-MOP and microglia display increased depletion of PIP₂ after exposure to physiologically relevant stimuli compared to P522 M-MOP and microglia.**

A–D   The level of PIP₂ in M-MOPS was examined by measuring immunofluorescence from images at set time points after exposure to 5 μg/ml anti-FcγRII/III (2.4G2), 50 ng/ml LPS or oligomers of 40 μM Aβ1–42 (A). The levels of PIP₂ were also measured using a plate reader after exposure to physiologically relevant stimuli in M-MOPS and primary mouse microglia (B). DAG level was measured using a live-cell assay in M-MOPS (and primary mouse microglia as a time course from immunofluorescence (C) and plate reader after exposure to physiologically relevant stimuli (D). Data show the mean ± SD of three independent experiments and were analysed by two-way ANOVA with Sidak's multiple comparison. *$P < 0.05$, **$P < 0.01$, ***$P < 0.001$ and ****$P < 0.0001$ (blue: P522 and red: R522). See also Appendix Fig S2.

**$Plcg2^{R522}$-expressing cells exhibit greater reduction of PIP₂ levels after PLCγ2 activation**

We next examined the change in levels of the PLCγ2 substrate PIP₂ during exposure to anti-FcγRII/III, LPS (50 ng/ml) or Aβ₁₋₄₂ oligomers (40 μM) by immunofluorescence (Fig 6A, Appendix Fig S6A

and B). These stimuli caused a significant reduction in PIP₂ levels in both macrophages and microglia. Cells expressing $Plcg2^{R522}$ demonstrated a greater reduction of PIP₂ compared to the $Plcg2^{P522}$ controls. This reduction in PIP₂ was prevented using the inhibitors edelfosine or U71322. This was also demonstrated in pre-incubation with LPS (50 ng/ml) and Aβ₁₋₄₂ oligomers (40 μM) but only over

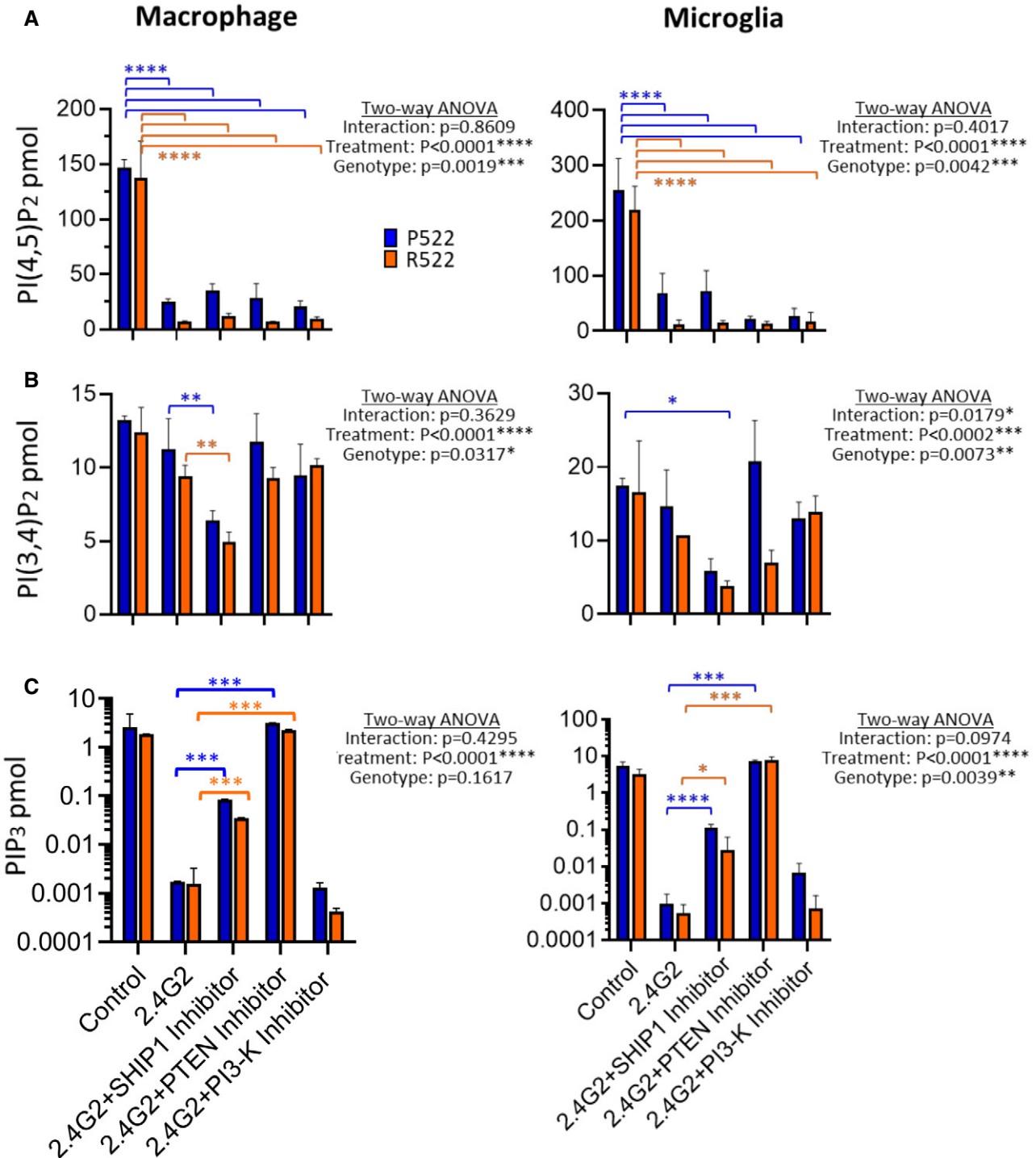

**Figure 7. Protective R522 M-MOP and microglia display increased depletion of PI(4,5)P₂ without corresponding compensation from other PIP species.**

A–C   A mass ELISA was used to detect specific PIP species after exposure to anti-FcγRII/III (2.4G2) with or without 3-a-aminocholestane (SHIP 1 inhibitor), SF1670 (PTEN inhibitor) and LY294002 (PI-3K inhibitor). In M-MOP cells (left panel) and primary microglia (right panel), PI(4,5)P₂ (A), PI(3,4)P₂ (B) and PIP₃ (C) were detected. All data show the mean ± SD of three independent experiments and were analysed by two-way ANOVA (log-transformed data were used in C) with Sidak's multiple comparison tests performed. *$P < 0.05$, **$P < 0.01$, ***$P < 0.001$ and ****$P < 0.0001$ (blue: P522 and red: R522).

longer time frames (Fig 6A and B). Using a DAG fluorescent sensor, we similarly saw that DAG levels increased after exposure to anti-FcγRII/III, LPS (50 ng/ml) and Aβ₁₋₄₂ oligomers (40 μM) (Fig 6C

and D). Similarly, both *Plcg2*^R522-expressing cells demonstrated a greater increase in DAG compared to the *Plcg2*^P522 controls (Fig 6C and D). This increase was prevented using the inhibitors edelfosine

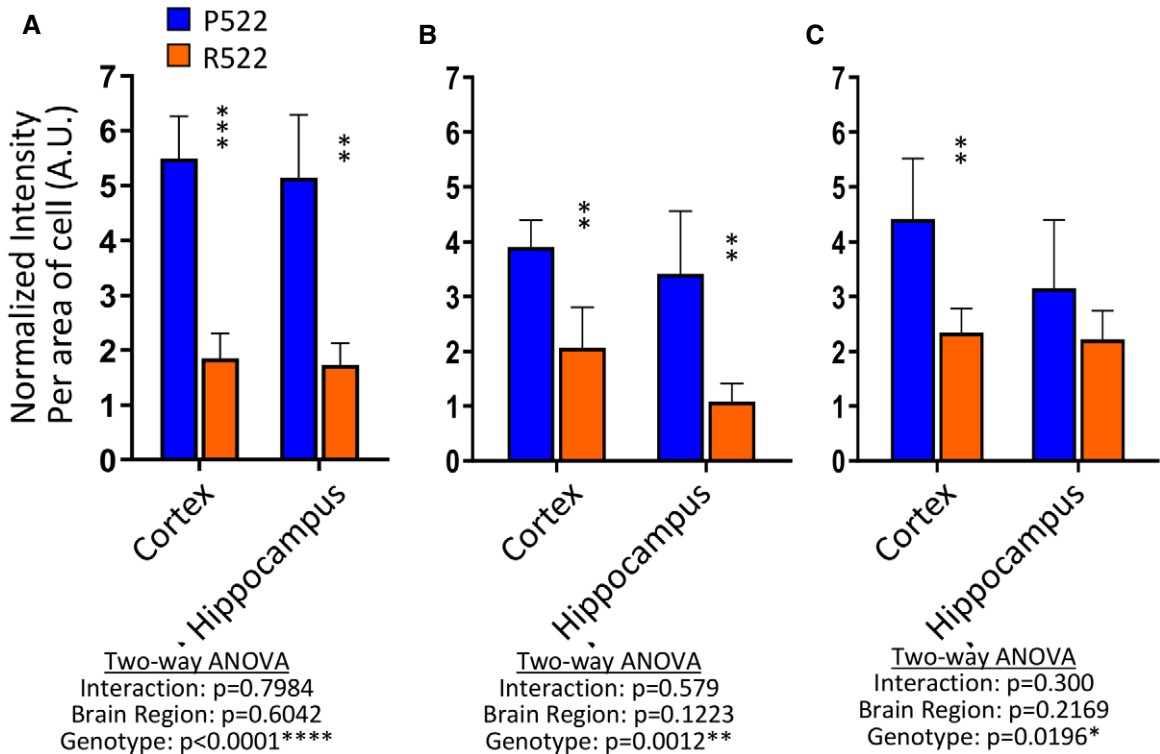

**Figure 8.** *In vivo* basal PIP$_2$ levels are decreased in microglia of *Plcg2*$^{R522}$ mice compared to *Plcg2*$^{P522}$ mice.

A–C   Iba1 was used as a marker for microglia, and PIP$_2$ levels were detected by measuring the intensity of florescence per cell. The average fluorescence was calculated above background in the cortex and hippocampus at 2 (A), 6 (B) and 9 (C) months. All data show the mean ± SD of three independent experiments and were analysed using two-way ANOVA with Sidak's multiple comparison tests performed. **$P < 0.01$ and ****$P < 0.0001$. See also Appendix Fig S7.

or U71322. Using a PI(4,5)P$_2$ mass ELISA to detect a more specific change in PIP species, we confirmed the reduction in PI(4,5)P$_2$ after exposure to anti-FcγRII/III and that there was a significantly greater decrease in the *Plcg2*$^{R522}$-expressing cells compared to the control (Fig 7A). Using 3-a-aminocholestane (SHIP1 inhibitor), LY294002 (PI3-K inhibitor) or SF1670 (PTEN inhibitor) did not prevent this reduction. Mass ELISA for PI(3,4)P$_2$ demonstrated no significant difference between the two variants after exposure to anti-FcγRII/III with or without the inhibitors (Fig 7B). Quantification of PI(3,4,5)P$_3$ by mass ELISA demonstrated no significant difference between the two variants after exposure to anti-FcγRII/III. But addition of either a SHIP1 inhibitor or a PTEN inhibitor with anti-FcγRII/III resulted in a significant reduction in PI(3,4,5)P$_3$ consumption (Fig 7C).

### *In vivo* basal levels of PIP$_2$ are reduced in cortical and hippocampal microglia in *Plcg2*$^{R522}$ mice

The cortex and hippocampus of 2-month-old *Plcg2*$^{R522}$ and *Plcg2*$^{P522}$ mice were examined for PIP$_2$ levels in microglia by immunofluorescent staining for PIP$_2$ and Iba1 (Fig 8, Appendix Fig S7). Results were calculated from three separate groups of mice per variant. Representative immunostaining is shown in Fig 8A. In both cortex and hippocampus (Fig 8A–C), the levels of PIP$_2$ in microglia were reduced by approximately 60% in the *Plcg2*$^{R522}$ mice compared to the *Plcg2*$^{P522}$ mice at 2 months indicating a reduced basal level of

PIP$_2$ in AD vulnerable regions, a pattern that was maintained at 9 months.

## Discussion

The heritability of AD is estimated to be between 58 and 79% (Gatz *et al*, 2006). This strong genetic influence in AD has provided novel insights into the mechanisms underlying disease and will hopefully lead to potential new approaches for drug design. One such opportunity is presented by the recent discovery of a protective rare coding mutation in *PLCG2* P522R (Sims *et al*, 2017). PLCγ2 is an enzyme amenable to drug development approaches and hence presents an opportunity for intervention in the development of AD. To aid rational drug design and understand how PLCγ2 contributes to the disease process, it is important to understand how rare coding variants affect cell activation in myeloid cells and microglia, the primary cell expressing PLCγ2 in the brain. To address the role of the protective R522 mutation in PLCγ2 and enable modelling in a physiologically relevant context, we have developed two novel models for its study: Kolf2 CRISPR gene-edited hiPSC and CRISPR gene-edited mice, both of which harbour the R522 variant, which is conserved across the species. The generation of these original models for the study of PLCγ2 and AD is an essential first step in understanding the underlying mechanisms that lead to disease.

We present evidence that the mutated R522 PLCγ2 protective variant is hyperfunctional and results in consistently increased intracellular $Ca^{2+}$ signalling in response to a variety of receptor-mediated stimuli in microglia and macrophages (Figs 2 and 3). This altered signalling is likely to directly contribute to protection against AD as it acts downstream of receptors, such as TREM2 and CSF1R, which have been previously implicated in AD. Furthermore, here we have shown that direct stimulation with DOPS-liposomes (a TREM2 ligand) is effective at triggering this hyper-response (Fig 3). An earlier study had indicated that the R522 variant may be hyperfunctional in COS7 and HEK293T expression systems (Magno *et al*, 2019). Importantly, our studies show that PLCγ2 with the R522 variant exhibits increased function in both human and mouse disease-relevant microglia when expressed at physiological normal levels, in the context of other PLC enzymes. Interestingly, here we show LPS and amyloid oligomers trigger calcium responses which are heightened in those cells with the R522 variant. Both these stimuli have been shown to signal through TLR4 (Calvo-Rodriguez *et al*, 2020) which is known to signal via PLCγ2 and PI(4,5)$P_2$ (Le *et al*, 2014).

Computational modelling suggests that this hyperfunction may occur via inhibition of an autoinhibitory domain within PLCγ2. Analysis revealed a change in flexibility and position of the SH2 domains which could feasibly modulate PLCγ2's ability to indirectly trigger intracellular $Ca^{2+}$release. Increased hydrogen bonding may have important implications for protein function as it shows a change in orientation and location of the mutated residue loop. This could have a considerable structural impact on a binding domain and it is possible that a change in the binding domain leads to the observed movement of the SH2 domain to which it is connected. The SH2 domain of this protein is believed to play a critical role in stabilizing the early signalling complex which is stimulated by BCR crosslinking, where the domain inhibits the active site of PLCγ2 (Wang *et al*, 2014). A known gain-of-function mutation (G993) in PLCγ2 increases external $Ca^{2+}$ entry causing an autoimmune and inflammatory response (Yu *et al*, 2005). The cSH2 domain interacts with residues surrounding the catalytic active site, resulting in the autoinhibition of the PLCγ enzymes. This inhibition is reversed by the interaction of cSH2 with a phosphorylated tyrosine residue, and this domain movement allows for access to the active site. PI(3,4,5)$P_3$ binds to the active site and recruits PLCγ isoforms to the plasma membrane, resulting in their activation. Studies in PLCß2 have shown residues in the region surrounding the loop, which connects two parts of the catalytic domain to limit substrate access to the active site. If the same is true of PLCγ2, then the R522 mutation may be causing a similar change to autoinhibition (Everett *et al*, 2009). These functional predictions help explain our observations of increased PLCγ2 activity within R522 variant microglia and macrophages.

Having observed increased PLCγ2-mediated IP3-depenent $Ca^{2+}$ signalling within our models and devised a possible mechanism underlying this hyperfunction at the protein level, we then investigated the effect of this hyperfunctionality on PLCγ2's substrate (PI(4,5)$P_2$) and product (DAG). PI(4,5)$P_2$ levels in R522 and P522 M-MOP were examined using fluorescein-conjugated anti-PI(4,5)$P_2$ following addition of either anti-FcγRII/III antibody (to stimulate PLCγ2), LPS or Aβ. Following addition of anti-FcγRII/III antibody, we found that the protective R522 variant resulted in a significantly greater drop in PI(4,5)$P_2$ when compared to P522 control at 30 s,

and this drop then appears to take longer to return to baseline. LPS and Aβ addition also appears to induce a greater drop in PI(4,5)$P_2$ in the R522- compared to P522-expressing cells (Fig 6A). Observations of reduced PI(4,5)$P_2$ are further supported in Fig 6B where anti-PI(4,5)$P_2$ staining (measured via plate reader assay) in M-MOP and microglia, following activation with anti-FcγRII/III antibody resulted in decreased PI(4,5)$P_2$ in the protective R522 compared with P522.

PLCγ2 is positioned downstream of important signalling receptors, such as TREM2 and CSF1R, and, as such, would be expected to be undergoing tonic activation. Having observed that stimulation of PLCγ2 leads to enhanced depletion of its substrate (PI(4,5)$P_2$) by the protective R522 variant, compared to the common P522 allele, we hypothesized that the R522 variant may be presented with a state of depressed substrate levels *in vivo*. We examined this by assessing PI(4,5)$P_2$ levels in microglia *in situ* in the brains of young P522- and R522-expressing mice. We observed a marked reduction in microglial PI(4,5)$P_2$ in the R522 mice compared with their controls (Fig 8) demonstrating that the impact of the protective variant was evident in healthy young animals.

Live-cell DAG assays demonstrated that protective R522 M-MOP and microglia produce more DAG than P522 controls when stimulated with anti-FcγRII/III antibody (Fig 6F–H). Moreover, DAG levels remained higher overtime in the R522 cells, which is consistent with a faster diffusion profile of DAG compared with PI(4,5)$P_2$) (Xu *et al*, 2017). Both direct addition and pre-treatment of M-MOP with LPS or Aβ again resulted in more DAG production in the R522-expressing cells compared with P522 cells, and a similar pattern of response was observed in the microglia.

PI(4,5)$P_2$, PI(3,4)$P_2$ and PIP3 were also measured via mass ELISA in P522 and R522 M-MOP and primary microglia, confirming reduced PI(4,5)$P_2$ in the protective R522 variant following PLCγ2 activation with anti-FcγRII/III antibody. This effect was not prevented by co-treatment with SHIP1, PI3K or PTEN inhibitors, demonstrating PI(4,5)$P_2$ reduction as an upstream event of these enzymes following PLCγ2 activation.

Taken together, our observations demonstrate increased PLCγ2 activity in R522 variant-expressing cells when compared to those expressing P522, which results in increased PI(4,5)$P_2$ consumption, and consequent DAG production. The increased PI(4,5)$P_2$ utilization associated with the R522 variant coupled with a slow recovery of PI(4,5)$P_2$ levels would lead to a depleted PI(4,5)$P_2$ pool compared with that in the common P522 variant cells. The delayed return to baseline, heightened tonic activation of PLCγ2 and consequent reduction in available PI(4,5)$P_2$ levels could limit further activation of PLCγ2 within this time, potentially preventing chronic enzyme activation or reducing sensitivity to further receptor-mediated stimulation. Aligned to these observations, mice expressing PLCγ2$^{R522}$ exhibited reduced microglial PI(4,5)$P_2$ *in vivo* (Fig 8 and Appendix Fig S7).

To understand how hyperactivity of PLCγ2 affects cell behaviour, we first investigated phagocytosis by the P522- and R522-expressing microglia and macrophages by measuring uptake of pHrodo-labelled *E. coli* and zymosan particles. Zymosan and *E. coli* are phagocytosed by distinct receptor-mediated mechanisms, with a notable role for dectin-1 (Brown *et al*, 2002), an ITAM-like containing signalling molecule that signals through PLCγ2 (Xu *et al*, 2009), in the clearance of zymosan. We observed an overall reduction in

phagocytosis of both *E. coli* and zymosan in all R522-expressing models when compared to the common P522 variant-expressing controls (Fig 4). This phenotype could be attributed to reduced PI(4,5)$P_2$ levels in the R522 variant cells and increased loading of these cells with PI(4,5)$P_2$ increased the uptake of these particles (Appendix Fig S3). PI(4,5)$P_2$ levels are critical to modulation of actin dynamics, and reduction of this lipid at the plasma membrane has been demonstrated to result in reduced phagocytosis in RAW 264.7 cells (Botelho *et al*, 2000b). This appears to be due to PI(4,5)$P_2$ roles in actin filament formation: a key step in phagocytosis (Scott *et al*, 2005a). PLCγ2 and PI(4,5)$P_2$ are also known to play an important role in phagosome formation and engulfment (Botelho *et al*, 2000a). In this process, the hydrolysis of PI(4,5)$P_2$ by PLCγ2 is required to occur in a carefully controlled manner. It is possible that the R522 variant, with its reduced autoinhibition, may unbalance this system with a consequence that the reduced PI(4,5)$P_2$ levels seen in R522 cells may affect the ability of cells to form phagosomes. We have demonstrated that inhibiting PLCγ2 or PI3K, which are both key components driving this system, do indeed reduce phagocytosis (Appendix Fig S4). However, inhibiting those components that inhibit these systems such as PTEN and SHIP1 can increase phagocytosis (Scott *et al*, 2005b). It is also interesting that increasing PI(4,5)$P_2$ in cells increases the rate of phagocytosis, especially for cells with the R522 variant, suggesting that the amount of PI(4,5)$P_2$ may be rate limiting (Appendix Fig S3). Interestingly, reduced phagocytosis has been detected in human IPSC microglia cell lines with a PLCG2 KO (Andreone *et al*, 2020). However, when low concentrations of bioparticles were used, there was no phagocytic deficiency in the R522 variant cells, with a suggestion that it may even be increased. Notably, using low doses of phagocytic target, another recent study observed a trend towards increased phagocytic uptake in R522 variant-expressing primary cells (Takalo *et al*, 2020). Taken together, this suggests that a sustained and greater challenge to the PLCγ2 signalling system leads to functional impairment. We suggest that this may be relevant in the *in vivo* context where continued pathway stimulation is expected to lead to a depletion of PIP$_2$ levels. However, more work is required to understand this.

Several recent findings suggest that increased phagocytosis may be an aggravating factor in AD (Nizami *et al*, 2019). For one, microglial activation and subsequent phagocytosis of stressed but still viable neurons has been postulated to enhance AD pathology (Gabandé-Rodríguez *et al*, 2020). Increased phagocytosis of apoptotic neurons has been observed in APOE4-overexpressing cells (Muth *et al*, 2019), and inhibiting microglial phagocytosis appears to prevent inflammatory neuronal death (Neher *et al*, 2011). Furthermore, increased synaptic pruning, which occurs via microglial phagocytosis of synapses, has also been implicated in AD (Rajendran & Paolicelli, 2018). Reduced phagocytic activity in the R522 variant microglia and subsequent reduction in phagocytosis of damaged but viable neurons and synapses may contribute to the protective effect assigned to the R522 variant of PLCγ2.

We also assessed endocytic clearance of 10 kDa dextran and soluble Aβ$_{1-42}$ oligomers, both of which are thought to be endocytosed by a variety of uptake methods (Wesén *et al*, 2017). Investigation using inhibitors of clathrin-independent (CIE) and clathrin-dependent endocytosis (CDE) suggested that both mechanisms are involved (Appendix Fig S2). We observed increased uptake in all R522 models when compared to the common P522 variant (Fig 5). Reasons for this increased uptake by cells expressing the protective variant are currently unclear, although several studies have previously implicated phosphoinositols as regulators of these forms of endocytic uptake (Brown *et al*, 2001). Both CIE and CDE require PI(4,5)$P_2$ but the requirement and the involvement of PLCγ2 is smaller when compared to phagocytosis (Donaldson, 2003; Nunes & Demaurex, 2010). However, other PIP species are of greater importance. Here, we show that increased PI(4,5)$P_2$ levels do not significantly increase endocytosis of amyloid or dextran (Appendix Fig S3), confirming current understanding from the literature. In addition, in contrast to what we see with phagocytosis, knockdown of PLCγ2 does not fully inhibit endocytosis of either dextran or amyloid (Appendix Fig S3). However, the decrease in endocytosis is greater in the cells with the R522 variant compared to the P522 variant. Taken together, this suggests that PLCγ2 only has a partial role in the uptake of these smaller particles (amyloid oligomers and dextran) which enter the cells by several endocytic systems (including CIE & CDE). Despite these observations, it is possible that the hyperfunction of the R522 variant is still directly responsible for the increased endocytic uptake in these cells. This could occur via the observed increased Ca$^{2+}$ response seen in cells with the R522 variant, as this is known to function upstream of several endocytic systems (Nunes & Demaurex, 2010).

Soluble Aβ has been demonstrated to perturb metabolic processes, induce the release of deleterious reactive compounds, reduce blood flow, inhibit angiogenesis and induce mitochondrial apoptotic activity (Watson *et al*, 2005). Enhanced internalization of Aβ$_{1-42}$ with R522 variant microglia when compared with common P522 could therefore reduce some of these toxic effects. Together, the results from our phagocytosis and endocytosis assays, which can be in part explained by observations of reduced PI(4,5)$P_2$, suggest mechanisms by which the R522 variant may protect against AD.

The above data suggest that the R522 variant results in a more activated microglial population. One clear area of further research would be to investigate the inflammatory component and cytokine release by these cells. Takalo *et al* (2020) recently showed that R522 variant macrophages treated with a high dose of LPS and IFNγ release a slightly increased amount of proinflammatory cytokines after 3 h compared to P522-expressing macrophages. The potential for an enhanced inflammatory response and its contribution to the protective role of the R522 in AD require more investigation.

In summary, we have shown a consistent increase in PLCγ2 enzymatic activation in novel human and mouse models due to the AD protective R522 mutation. This is associated with depletion of PI(4,5)$P_2$ *in vitro* after exposure to physiologically relevant stimuli. We also observe reduced phagocytic and increased endocytic clearance of multiple cargoes. We demonstrated that PLCγ2 and PI(4,5)$P_2$ has a varied role in the uptake of these different cargoes. *In vivo*, we have shown a basal decrease in mice with the R522 mutation in PLCγ2 compared to wild type.

These alterations in cell activities could directly impact on clearance of damaged cells and synapses *in vivo* both during disease and under homeostatic conditions where continual activation of PLCγ2 would be expected. Reduced PI(4,5)$P_2$ in the protective R522 variant-expressing microglia, whilst allowing for short bursts of PLCγ2 activity, may limit such longer term enzyme activation. The

generation of these original models for the study of PLCγ2 and AD has provided novel insight into the regulation of cell function and is an essential first step in understanding the underlying mechanisms that lead to disease.

# Materials and Methods

## Mice

Mice were maintained under specific-pathogen-free conditions with environmental enrichment as standard, and all experiments were conducted in accordance with UK Home Office (Animal Scientific Procedures Act 1986) and Institutional guidelines.

## Generation of Plcg2 knockin mice

Plcg2$^{R522}$ knockin mice (henceforth referred to as Plcg2$^{R522}$) were generated by CRISPR/Cas9-assisted gene targeting in B6(SJL)-Apoe$^{tm1.1(Apoe*4)Adiuj}$/J (JAX#27894) mice. Briefly, microinjection mix contained 100 ng of Cas9 and 50 ng of CRISPR guide (CCAA AATGCAGCTCCGTGGG) and 20 ng of the 81nucleotide donor: (GAGCTGTAATAAGCCCTTTCGGATGCTTGTGGCTCAGGACACTCG CCCCACGGAGCTGCATTTTGGGGAGAAATGGTTCCACA) was used to engineer a C to G mutation (underlined) at nucleotide 1,565 and change the CCC (proline) codon to CGC (arginine). This mutation disrupted the PAM sequence to prevent re-cutting. Founders were genotyped by PCR using forward primer 5'-GCGCTGCATGTTC CCTCTA-3' with reverse primer 5'-GGGCGTTACCAGAAGGAGAG-3' to generate a 375bp product, which was sequenced to identify the mutation. Positive founders were bred to in C57BL/6J (JAX #664) for two generations to breed away from the APOE4 allele. Once the line was established, genotyping was performed via Endpoint analysis using the amplifying primers: 5'-AATAAGCCCTTTCGGATGCT-3' and 3'-TCCGCACTGGTCCTACTCTC-5' with Wild Type Hex labelled probe 5'-Hex-AGGACACTCCCCCCACG - Black Hole Quencher 1–3' and Mutant 6-Fam labelled probe 5'-6-Fam-AGGACACTCGCCC CACG – Black Hole Quencher 1–3'. This model is available from the Jackson Laboratory as B6. Cg-Plcg2$^{em1Msasn}$/J (#29598). Control (Plcg2$^{P522}$) mice for the Plcg2$^{R522}$ mice were generated from the littermates of the founders.

## Mouse Plcg2-P522R variant genotyping

Ear biopsies were digested in 50 µl of mammalian lysis buffer (100 mM Tris–HCl pH 8.5, 5 mM EDTA, 0.2% (w/v) SDS, 200 mM NaCl) containing Proteinase K at 100 µg/ml and incubated at a temperature of 56°C whilst shaking at 1,200 rpm for 1 h in a 1.5-ml micro-centrifuge tube (Starlab Thermomixer HC S8012-0000). Once dissolved, the sample was heated for a further 30 min at 72°C to denature the Proteinase K. Once cooled to room temperature, the sample was diluted by the addition of 450 µl of nuclease free water. The sample was frozen at −20°C or immediately used for the qPCR. Genotyping was performed in a 10 µl reaction volume incorporating 5 µl of the TaqMan® Fast Advanced Master Mix (Applied Biosystems), with 0.9 µM of each amplifying primer (see above), 0.25 µM of each probe (see above) and 4.5 µl of the sample digest both from above.

## Generation and culture of conditionally immortalized macrophage precursor (MØP) cell lines

Conditionally immortalized macrophage precursor (MØP) was generated from bone marrow of wild-type Plcg2P522 and variant Plcg2R522 knockin mice and differentiated into macrophages with M-CSF (M-MØP) as previously described (Rosas et al, 2008, 2011). Both male and female lines were established, and the sex of the lines used in specific experiments is indicated in the main text. Briefly, CD117-enriched bone marrow cells (using CD117-biotin, BD Biosciences, and anti-biotin-MACS, Miltenyi Biotec) were infected with an oestrogen-dependent Hoxb8 encoding pMX-IPs retroviral vector for conditional immortalization and cultured in RPMI 1640 medium (Thermo Fisher Scientific) containing 10% (v/v) heat-inactivated foetal calf serum (FCS) (PAA Laboratories), 1% penicillin/streptomycin (Pen/Strep), β-oestradiol (1 µM) (Sigma-Aldrich), GM-CSF (10 ng/ml) (ImmunoTools) and puromycin (20 µM/ml) (Roth). After 10 days, puromycin was removed from the culture medium and the cells were passaged in the presence of β-oestradiol and GM-CSF. When required, Plcg2P522 and Plcg2P522 MØP cells were further differentiated to M-MØP by application of 20 ng/ml M-CSF for 4 days in the absence of β-oestradiol and GM-CSF. For all experiments, M-MØP cells were grown to 80% confluence and then starved of M-CSF for 4 h prior to performing assays.

## Generation of primary mouse microglial cultures

Primary neonate mouse microglia cultures were isolated from mixed glial cultures via a modified version of the previously described shaking technique (Tamashiro et al, 2012). Briefly, brains from Plcg2P522 and Plcg2R522 mice at P7-8 were collected and dissected into cortical and hippocampal sections in dissection media (HBSS with 0.1 M HEPES (Thermo Fisher Scientific), 1× penicillin/streptomycin, 2 mM L-Glutamine and 33 mM glucose (Sigma-Aldrich)). Cells were enzymatically (0.05% Trypsin) and mechanically dissociated and seeded in on Poly-L Lysine (1 mg/ml, Sigma-Aldrich) coated T75 flasks in DMEM (Thermo Fisher Scientific) containing 10% (v/v) heat-inactivated FCS, 1× penicillin/streptomycin, 2 mM L-Glutamine and M-CSF (10 ng/ml). After 10–14 days, cells were confluent and microglia were isolated from primary mixed cells via shaking at 200 rpm for 1.5 h at 37°C (Eppendorf Innova S44i). Microglia were plated in DMEM containing 10% heat-inactivated FCS, 1× penicillin/streptomycin, 2 mM L-Glutamine and M-CSF (10 ng/ml) on 8-well chamber slides or 96-well plates and left overnight to attach. Purity of cell cultures was verified by immunostaining with anti-glial fibrillary acidic protein (Abcam ab7260) and anti-Iba1 (Merck MABN92). For all experiments, primary microglia were grown to 80% confluence and then starved of M-CSF for 4 h prior to performing assays.

## Kolf2 hiPSC cell lines

Isogenic PLCG2 P522/P522 and PLCG2 R522/R522 hiPSC clones were derived from the male parent Kolf2 cell line, previously validated for CRISPR genome editing; for availability and full genetic profile of Kolf2 cells, see (http://www.hipsci.org/lines/#/lines/HPSI0114i-kolf_2). Kolf2 were grown on 10 µg/ml Geltrex coated culture plates (Nunc) in E8-Flex medium (Thermo Fisher). Colonies

were passaged 1:10 after dissociating to cell clumps using Cell Dissociation Buffer (Thermo Fisher).

## hiPSC culture and genome editing

Kolf2 hiPSC clones harbouring the PLCG2R522 variant were generated by CRISPR gene editing. Deskgen CRISPR design tools (no longer available) were used to select a guide RNA (CCAAAATGT AGTTCTGTAGG) and ss-oligonucleotide homology directed repair template (5′GTCAGGGTGAGACAGAAGGACCTGTCTAGTGATGCTG GGGTTTGGTCCAAGGCTTTCAGAAACCCCTCCTCTCTTTGCGGCCC AGGATATACGGCCGACAGAACTACATTTTGGGGAGAAATGGTTCC ACAAG3'). 2 nmol guideRNA and 20 nmol ATTO™ 550 labelled Alt-R® CRISPR-Cas9 tracrRNA were complexed in IDT buffer (all CRISPR reagents were purchased from Integrated DNA Technologies). The RNP complex was formed immediately prior to nucleofection by mixing the crRNA:tracrRNA complex with Alt-R® S.p. HiFi Cas9 Nuclease V3 (see Appendix Fig S8A).

hiPSC cultures pre-treated for 1 h with 10 μM Y-27632 were dissociated into single-cell suspension using Accutase (Sigma). $1 \times 10^6$ cells were resuspended in Amaxa P3 nucleofection buffer (Lonza), mixed with Cas9 RNP complex and 100 pmol of ssDNA oligonucleotide repair template and nucleofected using program CA137 on the Amaxa-4D nucleofector. After replating in E8-Flex medium containing 10 μM Y-27632 and overnight culture, nucleofected cells were harvested as a single-cell suspension using Accutase and the brightest Atto550 labelled cells (~5,000 cell) were harvested and replated onto a 10 μg/ml Geltrex coated 10 cm plate. Cells were fed with E8-Flex medium containing 10 μM Y-27632 and 100 U/ml penicillin/streptomycin for the first 3 days followed by E8-Flex only thereafter. Colonies derived from single cells were manually picked on day 7 into 96-well plates. 96-well plates were replica-plated by passaging with cell dissociation buffer allowing one plate to be used for DNA analysis. CRISPR screening was performed by PCR. Homology directed integration of the HDR template introduced a unique Eag1 restriction site enabling a simple and rapid 96-well plate-based PCR screen. Cells were lysed with DNA lysis buffer (QuickExtract, Lucigen) and PCR performed with the primers (forward TTTTCCCATACCCCTTCGGG and reverse AGTCATTGGGGAAGGTCTCG), and PCR products were digested with Eag1 and analysed by gel electrophoresis. PCR amplicons from candidate edited clones were sequence verified (Eurofins) and sequence analysed using CRISPR-ID software. The absence of editing at Deskgen predicted off-target sites was confirmed by PCR amplicon sequence analysis. Candidate edited clones were identified and expanded in E8-Flex medium.

## hiPSC microglial differentiation

hiPSCs were differentiated to microglia according to Haenseler *et al* (2017). Briefly, hiPSCs treated with 10 μM Y-27632 were dissociated using Accutase and resuspended in mTesR medium (Stemcell Technologies). Embryoid bodies (EB) were formed by aggregating 20,000 cells for 24 h in 20 μl hanging drops. EBs were collected and grown in suspension in 3TG differentiation medium (mTesR supplemented with 50 ng/ml BMP-4, 20 ng/ml SCF and 50 ng/ml VEGF-121). These EBs were cultured for 6–8 days (until initiation of EB cyst formation was observed) with half medium change after 2 days.

Approximately 20 cystic EBs were plated per well of tissue culture-treated 6-well plates and cultured in 3 ml haematopoietic medium (X-VIVO 15 (Lonza, LZBE02-060F) supplemented with 2 mM GlutaMax, 100 U/ml penicillin, 100 μg/ml streptomycin, 100 ng/ml M-CSF and 25 ng/ml IL-3) with half medium changes every 3 days (Appendix Fig S8B). After 2 weeks of differentiation, myeloid progenitor cells were harvested from the culture supernatant and plated on poly-D-lysine (0.1 mg/ml)/fibronectin (0.5 μg/cm²) coated culture plates in microglial differentiation medium (1:1 mix of ADF:ACM media; ADF is Advanced DMEM/F12 DMEM (Thermo Fisher), 2% B27 supplement (50× Thermo Fisher Scientific, 17504044), 2 mM GlutaMax, 100 U/ml penicillin and 100 μg/ml streptomycin). ACM is astrocyte conditioned ADF medium; hiPSC-derived astrocytes were differentiated as previously described (Serio *et al*, 2013). ACM was derived by pooling ADF media from confluent Nunc T500 triple layer flasks (Thermo Fisher). ACM aliquots were stored at −80oC. A single ACM batch was used for all experiments.

## Immunocytochemistry assessment of microglia differentiation from hiPSC

Cultured cells were washed once in PBS and fixed with 4% paraformaldehyde (10 min room temperature) followed by 3 PBS washes. Cells were permeabilized by treating with 100% ice-cold EtOH for 2 min at room temperature followed by 2 × PBS washes. Fixed cells were incubated with blocking buffer (3% (v/v) goat/chicken serum, 0.1% (v/v) Triton-X-100 in PBS) for 1 h at room temperature before overnight incubation at 4oC with primary antibodies diluted in blocking buffer. Primary antibodies used were anti-IBA-1 (1:100 Abcam AB5076), anti-Glut5 (1:100 R&D systems MAB1349), anti-TMEM119 (1:100 Abcam AB185333) and anti-P2RY12 (1:100 Abcam AB188968). Following overnight incubation, cells were three times with PBS prior to a 1-h incubation at room temperature in the dark with fluorescent secondary antibodies diluted in blocking solution. Secondary antibodies used were Alexa Fluor 594 chicken anti-goat IgG (Invitrogen A21468), Alexa Fluor 594 goat anti-rabbit IgG (Invitrogen A11037) and Alexa Fluor 488 goat anti-mouse IgG (Invitrogen A11029) all at 1:400 (Appendix Fig S8).

Coverslips were subsequently incubated with Hoechst 33258 (Thermo Fisher Scientific) at 1:5,000 in blocking buffer and mounted on microscope slides (Immu-Mount, Fisher, 9990402). Images were taken using a Leica DM18 confocal microscope (Appendix Fig S8C).

## Molecular dynamic modelling of PLCγ2 variants

P522 and R522 proteins were subjected to 300 ns of MD. MD was carried out, in triplicate, using the GROMACS package (Pronk *et al*, 2013) and the Amber03 force field (Case *et al*, 2005). The structures were boxed and solvated using the GROMCAS module. The molecule was placed in the centre of a cubic box and solvated using TIP3P charge water molecules. Neutralization of the system was carried out by adding an appropriate number of Cl– ions to the box in the place of water molecules. The particle mesh Ewald (PME) method was used to treat long-range electrostatic interactions and a 1.4 nm cut-off was applied to Lennard–Jones interactions. The MD simulations were all carried out in the NPT ensemble, with periodic boundary conditions, at a temperature of 310 K, and a pressure of 1 atm.

Each simulation was performed using a three-step process: steepest descent energy minimization with a tolerance of 1,000/KJ/nm; a pre-MD run (PR) with 25,000 steps at 0.002 per second per step making a total of 2,500 ps; and an MD stage run for a total of 300 ns. Root mean square deviation (RMSD) was monitored along with the total energy, pressure and volume of the simulation to check for stability.

Resulting structures were analysed for flexibility using the g_rmsf and hydrogen bonding using g_hbond (both GROMACS packages (Yang *et al*, 2014); all proteins were visualized for structural differences using VMD (Humphrey *et al*, 1996). Further to this, prediction of the functional effect and stability analysis was carried out using HoPE (Venselaar *et al*, 2010). HoPE analyses the impact of a mutation, taking into account structural impact, and contact such as possible hydrogen bonding and ionic interactions. Flexibility, rmsd, energy, pressure and volume distributions were tested for normality using the Anderson–Darling test. All were not normally distributed, and a Mann–Whitney *U*-test was used to determine any statistical differences between wild-type and mutated simulations as well as between simulation repeats. This is done using the wilcox test function in the R stats package.

### RT–qPCR

RNA was extracted using RNAeasy kit (Qiagen) following the manufacturer's protocol. All RNA was then quantified and normalized to 500 ng. SuperScript IV first-strand synthesis kit (Invitrogen) was used to create cDNA, and gene expression was analysed using the QuantStudio 5 (Applied Biosystems) and primer/probe assay for murine Plcg2 Mm01242530_m1 and Gapdh Mm99999915_g1 (Life Technologies).

### Western blotting

For MOP cells, cell lysates were prepared using RIPA lysis buffer solution (Santa Cruz sc-24948). Protein extracts (20 µg) were denatured at 70°C for 10 min and loaded on a Bolt 4-12 Bis-Tris plus gel (Invitrogen) and then transferred to nitrocellulose membrane (Novex). Membranes were blocked for 2 h in 5% (w/v) dried milk powder in TBST and then exposed overnight to anti-PLCG2 (Bio-Rad AHP2510) and anti-β-tubulin HRP conjugate (Cell Signaling 5346) at 4°C. Membranes were then washed with TBST and exposed to an anti-rabbit-HRP antibody (Jackson ImmunoResearch 711035152) for 2 h. The membrane was then developed in ECL (Thermo Fisher 32106) and imaged on a G:Box Syngene using GeneSys software. Bands were analysed using Image ProPremier software.

For hiPSC-derived myeloid precursors, whole-cell lysates were prepared using ice-cold RIPA buffer (Sigma) supplemented with mini protease inhibitor (Roche, 11836170001) and PhosStop reagent (Roche, 049068455001). Protein extracts (20 µg) were denatured in loading buffer at 70°C for 10 min and loaded into a 4-12% NuPA-GETM Bis-Tris plus gel (Life Tech, NP0336PK2) and ran in MOPS SDS running buffer at 165V for 45 min. Gels were transferred to nitrocellulose membranes that were then blocked with 5% (w/v) skimmed milk and probed with anti-PLCγ2 (Santa Cruz, sc-5283) overnight at 4°C or anti-α-tubulin (loading control, Abcam, ab7291) overnight at 4°C. Membrane were washed in PBS/0.1% Tween and incubated for 1 h at room temperature with IRDye 800CW Goat (polyclonal) Anti-Mouse IgG (H + L), Highly Cross Adsorbed

(1/10,000, LI-COR, 926-32210) prior to washing in PBS/0.1% Tween and imaging on a LI-COR Odyssey CLx imaging system. Bands were analysed and quantified using the LI-COR analysis software to confirm similar expression of the PLCγ2 variants (Appendix Fig S8D).

### GapmeR knockdown of mouse Plcg2

Plcg2 knockdown was achieved in M-MØP cells using an antisense LNA GapmeR produced against mouse Plcg2 (Qiagen). Cells were grown to 80% confluence (~2.4 × 105 cells per well) in 6-well plates. 3 µl of Lipofectamine RNAimax (Invitrogen) was diluted in 150 µl RPMI media and mixed with 150 pmol of siRNA in 150 µl RPMI media and left for 5 min. 250 µl of siRNA-lipid complex was added to cells in 3 ml RPMI. Cells were left for 48 h. Cells were then washed and used for $Ca^{2+}$ imaging (as described below), and knockdown of Plcg2 was confirmed with qPCR (as above). ΔΔCt values were used to calculate fold change and comparison of experimental and control samples used to measure knockdown. Negative and transfection control were performed in parallel.

### PIP$_2$ shuttle loading

Hyperloading of PIP$_2$ into mouse macrophage cells was achieved using a PIP$_2$ shuttle kit (Echelon Biosciences P-9045), as per the manufacturer's instructions. Briefly, carriers (P-9C3) were mixed with phosphoinositides (P-4516) (1:1) for 15 min at room temperature. The complex was diluted to 10 µM and added to cells in RPMI for 1 h. Medium was then removed and replaced with fresh RPMI. A complex made with fluorescent PIP2 was used for validation of the loading protocol but not for functional studies.

### Generation of 1,2-dioleoyl-sn-glycero-3-phospho-L-serine (DOPS) liposomes

The lipid phosphatidylserine, or alternatively 1,2-dioleoyl-sn-glycero-3-phospho-L-serine (DOPS), acts as a robust and reproducible anionic lipid ligand for TREM2 (Andreone *et al*, 2020). We therefore utilized DOPS-liposomes to activate TREM2 and access downstream $Ca^{2+}$ signalling within our cell culture models.

200 µl of 10 mg/ml chloroform solubilized DOPS (840035 Avanti Polar Lipids) was transferred to glass vials and dried for 2 h using a SpeedVac. Lipids were then hydrated with 250 µl of pre-chilled PBS and were sonicated in a water bath sonicator for 30 min. Every 10 min during this incubation tubes were removed from the sonicator and hand shaken. Next, a nanosizer mini sterile extruder (TT-0035-0010 TTScientific) was assembled using 200 nm polycarbonate membranes according to the manufacturer's instructions. The DOPS solution was transferred into a glass syringe, which was then inserted into one side of the polycarbonate membrane. Another glass syringe was inserted into the other side of the membrane. The mixture was then pushed back-and-forth through the membrane 31 times using the syringes. DOPS was then diluted to 4 mg/ml and stored at 4°C prior to use.

### Oligomerization of Aβ1–42

Amyloid beta (Aβ1–42) oligomers were produced in a similar manner to that described in (Ryan *et al*, 2010). Briefly, Aβ1–42

(A9810 Sigma) was suspended in hexafluoroisopropanol (HFIP, Sigma-Aldrich) to 1 mM. HFIP was then dried with a nitrogen stream and then lyophilized using a SpeedVac. Samples could then be stored for further use at −20°C. Peptide films were resuspended in DMSO to 5 mM. Samples were then sonicated for 10 min, diluted to 200 μM with ice-cold PBS + 0.05% SDS and vortexed for 30 s. Aggregation was allowed to proceed for 24 h at 4°C. The solution was further diluted with PBS to 100 μM and incubated for 2 weeks at 4°C.

### Single-cell imaging of Ca²⁺ signalling

Single-cell $Ca^{2+}$ imaging was performed using the ratiometric cytosolic $Ca^{2+}$ probe Fura-2 (Lee *et al*, 2015). Cells in chamber slides (Ibidi) were then loaded with 2 μM Fura2-AM (Abcam ab120873) solution at room temperature in the dark for 45 min. Cells were then washed and imaged in $Ca^{2+}$-free media, and EGTA was used to set the Fmin. Cells were imaged on an inverted Zeiss Colibri LED widefield fluorescence microscope with a high-speed monochrome charged coupled device Axiocam MRm camera and Axiovision 4.7 software. Primary mouse microglia and M-MØP cells were exposed at set time points to anti-FcγRII/III (2.4G2) antibody at 5 μg/ml (Stemcell Technologies), LPS (50 ng/ml) (Sigma-Aldrich) or oligomers of Aβ1–42 (0.5–40 μM). This was followed by ionomycin (2 μM) (Tocris) as a positive control and to set the Fmax. Cells were inhibited by pre-exposure for 2 h with edelfosine (10 μM) (Tocris) or U73122 (5 μM) (Sigma). hiPSC-derived microglia were exposed to anti-CD32 (Fisher 16-0329-81, 5 μg/ml). The ratio between the excitation at 360 and 380 nm was used to indicate intracellular $Ca^{2+}$ levels with regions of interest drawn over whole cells.

### Measurement of Ca²⁺ signalling by whole culture fluorimetry

Mouse macrophages and microglia were washed with HBSS and then loaded with 2 μM Fluo8-AM solution (Stratech) at room temperature in the dark for 1 h. Cells were washed and imaged in $Ca^{2+}$-free media and EGTA used as a negative control. Plates were warmed in a plate reader (Spectramax Gemini EM) or automated cell screening system (FLIPR Penta Molecular Devices) to 37°C and cells were exposed at set time points to anti-FcγRII/III, LPS, oligomers of Aβ1–42 or DOPS-liposomes (25 μg/ml), as above. This was followed by ionomycin (2 μM) as a positive control. Levels of fluorescence were detected at ex 490 nm/ em 520 nm (Hagen *et al*, 2012). Cells were inhibited by pre-exposure for 2 h with edelfosine (10 μM) (Tocris) or U73122 (5 μM) (Sigma) or xestospongin C (5 μM) (Abcam). Similarly, hiPSC-derived microglia were exposed to anti-CD32 (Fisher 16-0329-81, 5 μg/ml) or DOPS-liposomes (25 μg/ml).

### Assays of phagocytosis/endocytosis

hiPSC-derived microglia, primary microglia and M-MØP were prepared as described above. M-MØP cells were prepared with or without GapmeR KD of *Plcg2* or PIP₂ shuttle loading. In some assays, M-MØP cells were pre-exposed for 2 h with edelfosine (10 μM) (Tocris), U73122 (5 μM) (Sigma), xestospongin C (5 μM) (Abcam), 3-a-aminocholestane (20 μM, B-0341), LY294002 (10 μM,

B-0294) or SF1670 (5 μM, B-0350) to inhibit phosphoinositide metabolism. Similarly, in some assays, primary microglia and M-MØP were pre-exposed for 1 h to cytochalasin D (10 μM) (Sigma), for 30 min to Methyl-β-cyclodextrin (5 mM) (Sigma) or for 2 h to Chlorpromazine (10 μg/ml) (Sigma).

pHrodo-labelled-zymosan (Invitrogen, P35364) and *E. coli* (Invitrogen, P35361) bioparticles were diluted to 0.25 mg/ml, and pHrodo-labelled dextran (10,000MW, Invitrogen, P10361) was diluted to 0.1 mg/ml in live-cell imaging solution (Invitrogen, P35364) and then sonicated for 15 min using Bioruptor sonicator (Diagenode). In the dose titration studies, zymosan and *E. coli* were further diluted to 0.1, 0.05 and 0.01 mg/ml. Similarly, dextran was further diluted to 0.01 mg/ml.

On the day of the assay, hiPSC-derived microglia were washed with 100 μl of live-cell imaging solution and a phase image (20×) was taken of the plate using an IncuCyte Zoom Live-Cell Analysis System (Essen). The live-cell imaging solution was then removed, and a fresh 100 μl was added prior to the addition of 25 μl of the phagocytic/endocytic cargo. The hiPSC-derived microglia were put into the IncuCyte and imaged at 20× every 20 min for 4 h. An analysis pipeline was then set up on the IncuCyte which provided both percentage confluence (from the first phase scan) and total red object integrated intensity for each well at each time point. Total red object integrated intensity per well per time point was then divided by percentage confluence in order to obtain a normalized fluorescence reading per well. Similarly, the primary microglia and M-MØP cells were then imaged using an EVOS FL Auto 2 onstage incubator system (Thermo Fisher). Images at x40 from cells in the 8 well chamber slide were taken every 30 min until the 2 h mark, using the EVOS (transmitted light and RFP filter). Cells in the 96-well plate were read using the 544/590 filter on a BMG FLUOstar plate reader (BMG Labtech) to detect increased fluorescence per well. A BCA assay was run in parallel to normalize data to protein levels.

### Clearance of amyloid oligomers

hiPSC-derived microglia, primary microglia and M-MØP cells were prepared as described above. Cells were exposed to inhibitors as described above. Fluorescent amyloid oligomers (Eurogentech, AS-60479-01) were reconstituted to 0.1 mM by adding 1% $NH_4OH$ followed by PBS. This stock was then diluted in astrocyte conditioned media or RPMI (0.1–5 μM, as indicated) and was added to each well of the 96-well plate. After 2 h, media was removed and replaced with 0.2 mg/ml trypan blue for 2 min to quench extracellular fluorescence. The trypan blue was then removed, and cells were washed twice in live-cell imaging buffer. For iPSC-derived microglia, green fluorescence was measured in an IncuCyte Zoom Live-Cell Analysis System. Total green object integrated intensity per well was divided by percentage confluence in order to obtain a normalized fluorescence reading per well.

Primary microglia and M-MØP cells' green fluorescence was measured using the GFP filter on a BMG FLUOstar plate reader to detect increased fluorescence per well. Cells were imaged using the EVOS FL Auto 2 with the GFP filter. A BCA assay was run in parallel to normalize results to protein levels. Fluorescence was normalized further by taking the first reading at 0 mins as 0 arbitrary units (A.U.) for phagocytosis/endocytosis.

## Immunostaining of PIP2 *in vitro*

Microglia and M-MØP cells were prepared as described above. Cells were then exposed to anti-FcγRII/III (2.4G2) antibody at 5 μg/ml, LPS (50 ng/ml) or oligomers of Aβ1–42 (40 μM). Some cells were pre-incubated with LPS (50 ng/ml) or oligomers of Aβ1–42 (40 μM) for 1 h before a second dose was administered. Designated wells were inhibited by pre-exposure for 2 h with edelfosine (10 μM) or U73122 (5 μM). Other wells were pre-exposed to LPS (50 ng/ml) (Sigma-Aldrich) or oligomers of Aβ1–42 (40 μM) for 4 h. At set time points, cells were fixed (4% PFA, 10 min) and then washed with PBS twice. Cells were then permeabilized (0.05% saponin in PBS+Tween-20 (0.01%), 10 min) prior to washing with PBS and blocking (60 min, 10% BSA in PBS+Tween-20 (0.01%)). Cells were then exposed to fluorescein-conjugated anti-PI(4,5)P2 IgM (Z-G045 Echelon Biosciences, 1:200, 4 h). Cells were then washed three times with live-cell imaging solution, and green fluorescence was measured using the GFP filter on a BMG FLUOstar plate reader to detect increased fluorescence per well. Similarly, cells were imaged using the EVOS FL Auto 2 with the GFP filter at ×40 and intensity was measured with Image-Pro Premier software (Media Cybernetics). A BCA assay was run in parallel to normalize. Fluorescence was further normalized by taking the negative control unstained well at 0 mins as 0 A.U.

## Live-cell DAG assay *in vitro*

Briefly, cells were incubated at 37°C 5% $CO_2$ for 24 h with the BacMan sensor, sodium butyrate and receptor control. Cells were then washed with HBSS and left to rest for 30 min. Cells were exposed at set time points to anti-FcγRII/III (2.4G2) antibody at 5 μg/ml (Stemcell Technologies), LPS (50 ng/ml) (Sigma-Aldrich) or oligomers of Aβ1–42 (40 μM). Designated wells were inhibited by pre-exposure for 2 h with edelfosine (10 μM) (Tocris) or U73122 (5 μM) (Sigma). Other wells were pre-exposed to LPS (50 ng/ml) (Sigma-Aldrich) or oligomers of Aβ1–42 (40 μM) for 4 h. Green fluorescence was measured using the filter on a BMG Fluostar plate reader (BMG Labtech) to detect increased fluorescence per well. Similarly cells were imaged using the EVOS FL Auto 2 with the GFP filter at ×40, and intensity was measured with Image-Pro Premier software. A BCA assay was run in parallel to normalize. Fluorescence was further normalized by taking the negative control unstained well at 0 min as 0 A.U.

## PIP isolation from cells *in vitro*

Microglia and M-MØP cells were prepared as described above. Cells were then exposed to anti-FcγRII/III (2.4G2) antibody at 5 μg/ml or oligomers of Aβ1–42 (40 μM). Designated wells were inhibited by pre-exposure for 2 h with 3-a-aminocholestane (20 μM, B-0341), LY294002 (10 μM, B-0294) or SF1670 (5 μM, B-0350) (Echelon Biosciences). Cells were then exposed to ice-cold 0.5 M TCA (trichloroacetic acid T6399, Sigma) for 5 min. Cells were then scraped off and spun (1,720 *g*, 7 min, 4°C). The cell pellet was washed twice with 5% TCA/ 1 mM EDTA, vortexed for 30 s and centrifuged (1,720 *g*, 5 min). The pellet is then vortexed twice with MeOH: CHCl3 (2:1) for 10 min at room temperature and centrifuged (1,720 *g*, 5 min). The pellet is then vortexed with

MeOH: CHCl3:12 N HCl (80:40:1) for 25 min at room temperature and centrifuged at 1,720 *g* for 5 min. The pellet is then discarded, and CHCl3 and 0.1 N HCl are added. The tube is then centrifuged at 1,720 *g* for 5 min to separate organic and aqueous phases. Collect the organic phase into a new vial and dry in a vacuum dryer for 60 min.

## Quantification of PIP isoforms by mass ELISA

PIP isoforms were quantified from extracted lipids using PI(4,5)P2 Mass ELISA (K-4500), PI(3,4)P2 Mass ELISA (K-3800) and PIP3 Mass ELISA (K-2500s) from Echelon Biosciences using the manufacturer's instructions.

## Immunohistochemical detection of PIP2 in mouse brain

Wild-type Plcg2[P522] and variant Plcg2[R522] knockin mice were aged to 2, 6 or 9 months then trans-cardiac perfusion with 4% PFA was performed. The brains were removed and post-fixed with 4% PFA for 4 h at 4°C. Brains were then washed and cyroprotected with 30% sucrose until sunk. Brains were then embedded in OCT and cut using a CryoStar NX50 cryostat (Thermo Scientific) at 12 μm. Brain sections were mounted on super adhesive slides (Lieca) and stored at −80°C prior to fixing (2% PFA,10 min). Slides were then permeabilized (0.1% saponin, PBS-T (0.1%), 30 min) prior to quenching using 2 mg/ml ammonium chloride and blocking (10% BSA and 5% goat serum in PBST). Cells were then stained overnight at 4°C with anti-PI(4,5)P2 IgM (Z-G045 Echelon Biosciences 1:100) and anti-IBA1 (013-26471 Alphalabs 1:200). Sections were stained with DAPI in mounting media (H-1200 Vectorlabs) and sealed with nail varnish. Images were taken at x63 using a Cell Observer spinning disc confocal (Zeiss). Position in the brain section was found using anatomical markers, and images were taken at the primary somatosensory cortex and CA1 hippocampus. Images were analysed using Zeiss Blue 3.0 (Zeiss) and Image-Pro premier. Iba1 was used as a marker for microglia and used to create a region of interest. Inside that region of interest, the level of green fluorescence caused by staining of PIP2. Intensity of fluorescence per cell area was calculated, and background fluorescence was subtracted.

## Quantification and statistical analysis

Statistical analysis was conducted using Prism (GraphPad) or R and its inbuilt "stats" package. The details of the tests used and data representation are provided in the main text, in the results and figure legends sections.

# Data availability

This study includes no data deposited in external repositories.

**Expanded View** for this article is available online.

## Acknowledgements

This work was supported by the UK Dementia Research Institute at Cardiff, Dementia Platform UK and Centre for Ageing and Dementia Research. The Moondance Foundation, P.R.T. is also supported by a Wellcome Trust

Investigator Award (107964/Z/15/Z). G.E.M. is supported by a Ser Cymru II Fellowship, which is part funded by the European Regional Development Fund though the Welsh Government. J.W. is supported by Innovative Medicines Initiative (115736) and the Medical Research Council UK (HQR00720, MR/K013041/1). The work was in part supported by Eisai Inc. Additional costs were supported through an ARUK pump priming award (ARUK-NC2017-WAL), an ARUK Collaboration grant (ARUK-IRG2015-7 to E.L.E.) and MRC Partnership Award (MR/N013255/1 to N.D.A.). MODEL-AD is funded by U54 AG054345. The mouse CRISPR project was performed by The Jackson Laboratory Genetic Engineering Technologies group. This work was also funded in part by AG055104 (M.S. and G.R.H.).

## Author contributions

GEM, TP and EM contributed to acquisition and interpretation of data and to the drafting of the work. HMW, MAC, NE and ELC contribute to the acquisition and analysis of data. MS and GRH contributed to the design of the work and interpretation of the data. RS, EL-E, JW, NDA and PRT contributed to conception and design of the work. EL-E, JW, NDA and PRT contributed to the drafting of the work. All authors approved the submission.

## Conflict of interest

Aspects of the work were funded by Eisai Inc.

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
