## [Review Process File · The EMBO Journal]

The Alzheimer's disease protective PLCy2 R522 hypermorph depletes PIP2 and alters endocytosis.

Emily Maguire, Georgina Menzies, Thomas Phillips, Mike Sasner, Harriet Williams, Magdalena Czubala, Neil Evans, Emma Cope, Rebecca Sims, Gareth Howell, Emyr Lloyd-Evans, Julie Williams, Nicholas Allen, and Philip Taylor

DOI: [10.15252/embj.2020105603](https://doi.org/10.15252/embj.2020105603)

Corresponding author(s): Philip Taylor (taylorpr@cardiff.ac.uk) , Nicholas Allen (AllenND@cardiff.ac.uk)

Review Timeline:

Submission Date:	12th May 20
Editorial Decision:	6th Jul 20
Revision Received:	17th Mar 21
Editorial Decision:	26th Apr 21
Revision Received:	6th May 21
Accepted:	4th Jun 21

Editor: Karin Dumstrei

Transaction Report:

Dear Phil,

Thank you for submitting your manuscript to The EMBO Journal. I am sorry for the slight delay in getting back to you with a decision, but I have now received the full input on your manuscript.

As you can see from the comments below the referees find the analysis interesting, but also indicate that much further work would be needed in order to support the key conclusions. Should you be able and willing to take on major revisions then we would be willing to look at a revised version. However, I should note that we would need support from the referees to move forward with the manuscript.

I think it would be helpful to discuss the revisions further and I happy to do so via phone or video. I am away from the office this week, but we can do so next week if that works for you. Regarding the point raised regarding novelty (referee #1), as the related study came out while your study was under review here the issue of novelty is not an issue for us. However, the related work should be cited and discussed.

Thank you for the opportunity to consider your work for publication. I look forward to your revision.

Yours sincerely,

Karin

Karin Dumstrei, PhD
Senior Editor
The EMBO Journal

When assembling figures, please refer to our figure preparation guideline in order to ensure proper formatting and readability in print as well as on screen:
<http://bit.ly/EMBOPressFigurePreparationGuideline>

Further information is available in our Guide For Authors:

The revision must be submitted online within 90 days; please click on the link below to submit the revision online before 4th Oct 2020.

Referee #1:

In the current manuscript, Maguire et al., describe functional alterations in the Alzheimer Disease associated rare coding variant of the PLGC2 in mouse and human microglia. Here, authors show that PLCy2-R522 isoform results in a hyperactivation of the enzyme resulting in increased Ca²⁺ calcium levels, decreased enzyme substrates and increased metabolites. Besides, Maguire and colleagues also describe alterations in phagocytosis, hypothesizing that this pathway may play an important role in the disease. In the context of AD, this study is interesting since PLCy2-R522 variant is protective) and little is known regarding its function. Despite gathering an important amount of data, the repetition of the same experiments in multiple set ups (immortalized cell lines, mouse primary microglia and human iPSCs) is redundant. Besides, recent studies published (see <https://doi.org/10.1038/s41593-020-0650-6>) decreases the novelty of the findings. Other major comments are listed below:

- Maguire et al., show that PLCy2-R522 isoforms display an impaired phagocytosis compared to WT isoforms (Figure 4). However, in figure 5, they show that PLCy2-R522 isoforms have an increased endocytic capacity. I am a bit puzzled bit these findings. Phagocytosis is a subtype of endocytosis, how can authors explain that depending on the stimuli they apply, there are differences in the engulfment ability of the microglia?

- The newest studies on the role of PLCy2 (see <https://doi.org/10.1038/s41593-020-0650-6>) have shown that PLCy2 KO hiPSCs-derived microglia have an impaired phagocytic capacity. Here, authors postulate that PLCy2-R522 variants increase their enzymatic activity. Hence, wouldn't one

expect that PLCy2-R522 variants have enhanced phagocytosis? Are therefore PLCy2-R522 variants leading to a gain-of-function or loss-of-function? Even though authors nicely show that PLCy2-R522 increases its metabolic activity in the PIP2 pathway, does it mean that PLCy2 regulates phagocytosis via other pathways (independently of PIP2 metabolism)? Are the effects depending on the upstream receptor that activates PLCy?

- In line with the previous comment, authors also discuss that PLCy2 acts downstream TREM2, a well-known AD risk factor. However, experiments studying the interaction of PLCy2-R522 with TREM2 are missing. It would be of great interest to show how stimulation of TREM2 affects PLCy2 signaling depending on its isoforms.
- Authors stimulate microglia with LPS (50ng/ml) and oAb (40µM). How were these concentrations chosen? LPS concentration is within the range of currently in vitro studies. However, 40uM of oAb seems too much, especially when it has been shown that already 2.5 uM stimulation in primary neurons results in cell death (for instance, see <https://doi.org/10.1186/s13041-016-0284-5>).
- In the same vein, LPS and oAb activate the TLR4 receptor. PLCg2 has been shown to interact with TLR4, which is interesting. However, no reference is made to TLR4 and it is not mentioned that the previous experiment acts via TLR4. This makes this experiment a bit odd. Further discussion of this mechanism will facilitate the reading and strengthen the findings.
- As authors discuss in the introduction, hyperactivation of PLCy2 signaling may result in autoimmune diseases. Here, authors describe that PLCy2-R522 leads to hyper function of the enzyme. Would that result in an increased inflammatory reaction? It would be important to assess the levels of some pro- and anti-inflammatory cytokines upon stimulation to understand if this hyperactivation is driving to an enhanced inflammatory response and the consequences that it might have in the disease. Why is this mutant not causing autoimmune disease?

Minor:

- Discussion of their data in context of the newest findings regarding PLCy2 activity is needed.
- In figure 7 authors state that the experiments were done in 3 independent replicates. However, in panel C in some of the bars the SD are missing.

Referee #2:

A single amino acid mutation (P522R) in PLCG2 reduces the risk of Alzheimer's disease (AD). While the genetic causality has been well established, the functional consequences of the point mutation and its mechanistic link to AD have remained largely unclear and are currently a hot topic. For example, a few days ago (after this manuscript was submitted) another paper was published (Andreone et al. Nature Neuroscience 2020) that established the function of PLCG2 using knock-out microglial cells. The manuscript by Maguire and colleagues goes an important step further and establishes how the P522R mutation alters PLCG2 function using different cell models and finally mice. This study is novel, timely and a major advance for a future comprehensive preclinical validation of PLCG2 as an AD drug target. Yet, there are a number of points that require attention to improve the manuscript. This includes mechanistic studies (is the functional change downstream of TREM2? Why are different endocytic/phagocytic cargos affected differentially by the mutation) and a more detailed description/explanation of several experiments.

Major points:

1. The authors describe that PLCG2 acts downstream of TREM2, a master regulator of microglial function, but do not provide evidence whether this is also true for the functional changes of the R522 mutant. Given that some microglial functions are TREM2-independent (see for example Andreone et al. Nature Neuroscience 2020), it is important to test TREM2-dependency, for example using the calcium or phagocytosis assays.
2. Figures 4 and 5: Phagocytosis of bacteria is reduced with the mutant, while endocytosis of Abeta and dextran is enhanced. The authors need to provide a mechanistic explanation for this discrepancy. And can they be sure that one process is phagocytosis, while the other one is endocytosis? If this is the real reason for the discrepancy, they should demonstrate this, for example using inhibitors of one or the other pathway.
3. Fig. 2C: Describe this experiment in more detail. Are there different kinetics (rapid rise for P522 and slow rise for P522)? Or is the trace not representative? And why is the increase so much stronger compared to the data in panels A and B?
4. Fig. 3D: The authors need to better explain/speculate - at least in the discussion - how the increased Ab-induced calcium signaling may mechanistically be coupled to the increased Ab uptake in the phagocytosis assay.
5. Fig. 6B: Describe in more detail why there is a difference in PIP2 levels depending on whether LPS and Ab42 were preincubated or not. And why is the same not seen in panel 6D for macrophages versus microglia?
6. Fig. S1A: describe this panel in more detail. It is unclear what the x- and y-axes refer to and how this panel distinguishes between phenotypes.

Minor points:

7. Figure 1C: provide a magnification of the structure on the right, where you indicate the position of amino acid 522. The magnification will also allow a better visualization of the structural changes in the center of the structure.
8. Fig. 3B. Indicate whether RNA or protein levels are shown.

Referee #3:

In the present manuscript „The R522 Alzheimer's protective PLCy2 hypermorph depletes substrate and alters cell function" Maguire and colleagues demonstrate that cells with the R522 mutation show a hyperfunctionality which results in enhanced calcium store release in response to Fc-receptor ligation or A β oligomers. Furthermore, the authors show that the R522 variant causes reduced basal PIP2 levels and increased PIP2 depletion upon stimulation. Finally, the PLCy2-R522 variant lead to impaired phagocytosis and enhanced endocytosis.

The paper shows some interesting results, but presents a number of drawbacks. Some of the conclusions are not supported by the results shown (see specific comments below). In general, the manuscript is difficult to read and to understand because simple structure is missing e.g.: What are the main questions? What experiments were performed in order to answer the questions, what are the main results and what are the conclusions? Although the authors would like to make a link to Alzheimer's disease especially in the second part of their study when they treated the cells with A β oligomers, this part of the study would definitively improve with in vivo experiments by crossing the

PLcg2R522 mice with an AD mouse model.

Thus, in its present form the paper is not acceptable for publication and would require major revision. It is unfortunate that the current manuscript is not sufficient to provide a convincing data set that may support the authors' claim as listed below.

Specific points:

- 1) In the present study mice were housed in an environmental enrichment (EE), however it has been shown that EE can have an impact on microglia (Xu H et al. 2016 J Neurosci; Xu H et al. 2018 EMBO Molecular Medicine). How does housing the mice in an enriched environment affect the results?
- 2) In general, the quality of the Figures is lousy/poor and consequently the authors over-interpret their data. e.g. Figure 5 measurement of dextran in iPSC microglia is missing and needs to be shown, in Figure 6 there are no images for immunofluorescence shown only quantification is provided; images in supplementary Figure 2 are not convincing at all and images of A β are completely missing as well as images of microglia (only macrophages are shown). In Figure 8 only the intensity was measured for PIP2 levels and a real quantification is missing and needs to be done. From the images, it is questionable, if the PIP2 signal is really specific since in Supplementary Figure 2 it looks completely different.....Also, only images from the cortex are displayed but intensity was also measured in the hippocampus however representative images are absent here. Finally, morphological changes need to be detected in high resolution images in 3D and reconstructed with the image analysis software IMARIS.
- 3) Figure 4: iPSC microglia behave differently in C (E. coli) compared to F (Zymosan). There was already a significant difference after 1 hour detectable in F whereas in C only at the later time points 3+4 hours. Please explain.
- 4) Figure 5: In C only a graph is shown and no curve like in A and B. Why is this? Endocytosis in IPSC microglia should be measured over time the same way than in A and B.
- 5) Figure 8: the signal for PIP2 looks completely different than the PIP2 staining in the cells (Supplementary Figure 2). Why is this?
- 6) Supplementary Figure 3 and Supplementary Figure 4 are not discussed in the results section at all.
- 7) The discussion is too unfocussed. It is not clear what the main findings are and which ones are worth to discuss.

In summary, the results maybe of potential interest, but the data and interpretation has to be bulletproof and this is not the case.

Response to reviewers' comments

We appreciate the time and effort that you and the reviewers have dedicated to providing your valuable feedback on my manuscript. We are grateful to the reviewers for their insightful comments and suggestions to our paper. Since receiving these comments, we have worked to answer and incorporate as many of your suggestions as possible to the manuscript. In addition, since submission of the paper there was a new crystal structure released of the PLCG1 protein ([10.7554/eLife.51700](https://doi.org/10.7554/eLife.51700)). We have used this to make an updated model and subsequently updated the modelling methods and results following this. We believe the results to be clearer using this new model.

Throughout this document:

- **Reviewers comments will be shown in bold.**
- Our responses will be shown in standard script.
- *Truncated quotes from the new document will be italicized and coloured blue to facilitate navigation of the revised manuscript.*

When referring to figures in this document this refers to figure numbers in the revised version unless stated otherwise.

Reviewer 1

1 Despite gathering an important amount of data, the repetition of the same experiments in multiple set ups (immortalized cell lines, mouse primary microglia and human iPSCs) is redundant.

We do not agree that the data is redundant. The replication of data across species gives confidence in a conservation of function and validation of the respective models for use in PLCy2 studies: the use of iPSC-derived microglia as a surrogate for primary human microglia and validating the mouse model (the first time this specific model is reported) as one that provides relevant data for the study of PLCy2 variants. Similarly, the validation of conditionally-immortalised macrophage precursors as a macrophage model, provides a highly tractable and rapid model for the study of PLCy2.

2 Besides, recent studies published (see <https://doi.org/10.1038/s41593-020-0650-6>) decreases the novelty of the findings

The work by Andereone *et al.*, was published following the submission of our manuscript, and whilst this study clarifies the role of PLCy2 function in the context of TREM2 it does not examine the R522 variant or provide insight into how it may alter cellular function and the development of AD. Our work, which demonstrates a hyper-functional PLCy2 response within the protective variant, and in turn demonstrates reduced overall substrate alongside functional changes, could not be inferred from Andereone *et al.*

3 Maguire *et al.*, show that PLCy2-R522 isoforms display an impaired phagocytosis compared to WT isoforms (Figure 4). However, in figure 5, they show that PLCy2-R522 isoforms have an increased endocytic capacity. I am a bit puzzled by these findings. Phagocytosis is a subtype of endocytosis, how can authors explain that depending on the stimuli they apply, there are differences in the engulfment ability of the microglia?

Phagocytosis is a subtype of endocytosis, but this does not mean that the mechanisms of uptake by all forms of endocytosis are the same. There are multiple different uptake mechanisms utilized by microglia and macrophages. These functions each have distinct, although often overlapping, cellular mechanisms. The main pathway utilized for uptake of both zymosan and *E. coli* is receptor-mediated

phagocytosis, with well-characterised and distinct receptor profiles recognising the two targets. Of note, one of the main receptors for zymosan on macrophages is dectin-1 (PMID: 12163569), an ITAM-like containing signalling molecule that signals through PLC γ 2 (PMID: 19136564). Dextran and amyloid, however, are believed to be taken into cells via alternate forms of endocytosis. To illustrate this, we have included data with a variety of inhibitors with differential effects on the uptake of the various cargoes (new Supplementary figure 2, please also see below). Edits to the text to account for the new data are also shown below (lines 456-471).

Cytochalasin D, which inhibits actin cytoskeleton remodelling, was able to inhibit all forms of uptake as expected. In contrast, chlorpromazine, an inhibitor of clathrin-mediated endocytosis, was able to almost completely abolish dextran/amyloid uptake whilst having minimal effect on uptake of zymosan/*E.coli*. In addition, methyl- β -cyclodextrin, which inhibits clathrin-independent endocytosis, was unable to inhibit uptake of zymosan/*E. coli* whilst significantly affecting dextran/amyloid uptake.

Different particles are taken in via different subtypes of endocytosis, which each utilize different cellular machinery and co-factors. In addition, the use of distinct receptor-mediated recognition events for different cargoes with differential dependence on PLC γ 2, will also effect the outcome of these uptake assays. This means that different subtypes can be affected differently within genetically altered microglia such as the R522.

Results (lines 456-471):

"Inhibition of actin cytoskeleton remodelling using Cytochalasin D effectively reduced uptake of all examined cargos by >90% For all these inhibitors the effect was equal on cells with both variants."

4 The newest studies on the role of PLC γ 2 (see <https://doi.org/10.1038/s41593-020-0650-6>) have shown that PLC γ 2 KO hiPSCs-derived microglia have an impaired phagocytic capacity. Here, authors postulate that PLC γ 2-R522 variants increase their enzymatic activity. Hence, wouldn't one expect that PLC γ 2-R522 variants have enhanced phagocytosis? Are therefore PLC γ 2-R522 variants leading to a gain-of-function or loss-of-function? Even though authors nicely show that PLC γ 2-R522 increases its metabolic activity in the PIP2 pathway, does it mean that PLC γ 2 regulates phagocytosis via other pathways (independently of PIP2 metabolism)? Are the effects depending on the upstream receptor that activates PLC γ ?

We believe that our results and those of Andreone et al. are not mutually exclusive. Where Andreone et al., which show impaired phagocytic capacity in PLC γ 2 KO human iPSC-derived microglia we showed reduction in phagocytic activity with the R522 variant which demonstrates increased enzymatic activity (Figure 2). Of note, and fully aligned to the studies of Alderone et al., we also see impaired phagocytic uptake with the loss of PLC γ 2 via GapmeR inhibition (now added in Supplementary Figure 3A and B).

One might expect that an over-expressing mutant would show opposite results to the KO. However, as is demonstrated in Figure 6, increased PLC γ 2 activity in the R522 mouse microglia and macrophages results in increased PI(4,5)P $_2$ breakdown. This increased breakdown results in reduced levels of PI(4,5)P $_2$ both *in vitro* (Figure 7) and *in vivo* (Figure 8) within the *Plcg2*-R522 expressing mouse. In this way, we postulate that hyper-activity of the PLC γ 2-R522 enzyme, particularly in the context of higher or sustained demand, may result in a quasi-loss-of-function due to substrate limitation. PI(4,5)P $_2$ acts as a crucial co-factor for actin polymerization during phagocytosis, meaning that reduced PI(4,5)P $_2$ could explain the reduced phagocytosis of zymosan and *E. coli* observed in our studies. This hypothesis is further probed in the new supplementary figure 3E, F, G, & H (see below). These figures show how supplementation of PI(4,5)P $_2$, using a PIP $_2$ shuttle, was able to increase *E. coli* uptake in both P522 and R522 macrophages and zymosan uptake in *Plcg2*-R522 expressing cells. PI(4,5)P $_2$ add-back however had no effect on either dextran or amyloid uptake, which is consistent with why these pathways are not impaired in the PI(4,5)P $_2$ depleted R522 cells.

Edited text from the manuscript explaining results in Supplementary figure 3 can be found in the revised manuscript.

Results (lines 474-488):

We next investigated the role of phosphoinositide metabolism on phagocytic and endocytic uptake within our models. Inhibition of PTEN increased uptake of E.coli (Supplementary Figure 5A,B) while SHIP1 inhibition increased uptake of E.coli but did not significantly increase dextran uptake.

5 In line with the previous comment, authors also discuss that PLC γ 2 acts downstream TREM2, a well-known AD risk factor. However, experiments studying the interaction of PLC γ 2-R522 with TREM2 are missing. It would be of great interest to show how stimulation of TREM2 affects PLC γ 2 signaling depending on its isoforms.

We agree that given the close interaction between PLC γ 2 and TREM2, experiments investigating TREM2 activity within our models would be of value to our manuscript. For this reason, we have included new experiments in Figure 3 C, D, and E (see below) which analyse the resulting Ca²⁺ response in P522 vs R522 macrophages and microglia (mouse and human) following TREM2 activation using phosphatidyl-serine containing liposomes, which are known ligands of TREM2 (PMID: 32514138). These experiments demonstrate increased cytoplasmic Ca²⁺ increase by cells harbouring the PLC γ 2-R522 variant when stimulated with the liposomes, consistent with PLC γ 2-R522 being hyperfunctional downstream of TREM2. This hypothesis is supported by observations in Andrenone *et al.*, which demonstrated how the addition of such liposomes primarily exerted their downstream effects via PLC γ 2. Effects of the R522 mutation on TREM2 signalling, given the aforementioned importance of TREM2 as an AD risk gene, has significant implications for disease pathology.

The following changes were made to the manuscript:

Methods (lines 157-173):

Generation of 1,2-dioleoyl-sn-glycero-3-phospho-L-serine (DOPS) liposomes

The lipid phosphatidylserine, or alternatively 1,2-dioleoyl-sn-glycero-3-phospho-L-serine (DOPS), acts as an anionic lipid ligand for TREM2 (Andreone et al, 2020). We therefore utilized DOPS-liposomes to activate TREM2 and access downstream Ca²⁺ signalling within our cell culture models.....Another glass syringe was inserted into the other side of the membrane. The mixture was then pushed back-and-forth through the membrane 31 times using the syringes. DOPS was then diluted to 4 mg/ml and stored at 4 °C prior to use.

Methods (lines 193-205):

Measurement of Ca²⁺-signalling by whole culture fluorimetry

Mouse macrophages and microglia were washed with HBSS then loaded with 2 μ M Fluo8-AM solution (Strattech) at room temperature in the dark for 1 hour..... Similarly, hiPSC-derived microglia were exposed to anti-CD32 (Fisher 16-0329-81, 5 μ g/ml) or DOPS-liposomes (25 μ g/ml)..

Results (lines 420-423):

DOPS-liposomes results

Using DOPS-Liposomes to activate TREM2, we found increased cytosolic Ca²⁺ release in all three examined models with the R522 variant compared to those with the P522 variant (Figure 3C-E)

Discussion (lines 549-550):

DOPS-liposomes discussion

We have shown that direct stimulation with DOPS-liposomes (a TREM2 ligand) is effective at triggering this hyper response (Fig. 3).

6 Authors stimulate microglia with LPS (50ng/ml) and oAb (40µM). How were these concentrations chosen? LPS concentration is within the range of currently in vitro studies. However, 40uM of oAb seems too much, especially when it has been shown that already 2.5 uM stimulation in primary neurons results in cell death (for instance, see <https://doi.org/10.1186/s13041-016-0284-5>).

We agree with the reviewer that the 40µM dose of oAb was a relatively high dose with respect to other studies. As such we have now shown the calcium assays with a 0.5 µM dose of oAb to match the endocytosis assays we performed. The original data has been moved to supplementary figure 1J. The 0.5 µM dose of oAb does not elicit a strong Ca²⁺ signal, making it difficult to detect a significant difference between the PLCγ2 variants and necessitating the use of higher doses in these very short-term exposure experiments to increase the signal. These short term microglial-stimulation experiments are not comparable to the longer term neuronal death studies mentioned by the reviewer and conducted for a discrete purpose.

7 In the same vein, LPS and oAb activate the TLR4 receptor. PLCγ2 has been shown to interact with TLR4, which is interesting. However, no reference is made to TLR4 and it is not mentioned that the previous experiment acts via TLR4. This makes this experiment a bit odd. Further discussion of this mechanism will facilitate the reading and strengthen the findings.

We agree with this reviewer that when evaluating the cellular response to LPS and oAb, it is important to discuss how it arises via TLR4 signalling. For this reason, we have updated our discussion accordingly and would like to thank the reviewer for bringing it to our attention.

Discussion (lines 554-558).

Interestingly, here we show LPS and amyloid oligomers trigger a calcium response which is heightened in those cells with the R522 variant. Both these stimuli have been shown to signal through TLR4 (Calvo-Rodriguez et al, 2020) which is known to signal via PLCγ2 and PI(4,5)P2 (Le et al, 2014).

8 As authors discuss in the introduction, hyperactivation of PLCγ2 signaling may result in autoimmune diseases. Here, authors describe that PLCγ2-R522 leads to hyper function of the enzyme. Would that result in an increased inflammatory reaction? It would important to assess the levels of some pro- and anti-inflammatory cytokines upon stimulation to understand if these hyperactivation is driving to an enhanced inflammatory response and the consequences that it might have in the disease. Why is this mutant not causing autoimmune disease?

During revision of this manuscript an additional study has been published that shows an immediate and enhanced inflammatory cytokine response to LPS (Takalo et al., 2020, PMID: 32917267). As to why is this mutant is not causing overt autoimmune diseases, it could simply be that a more modest enhancement of function enzymatic is insufficient, but this area will require more specific investigation.

9 Discussion of their data in context of the newest findings regarding PLCγ2 activity is needed.

We have now added discussion of the manuscripts published whilst ours was in submission.

Discussion (lines 651-652)

Interestingly reduced phagocytosis has been detected in human iPSC microglia cell lines with a PLCG2 KO (Andreone et al 2020).

Discussion (lines 655-663)

Notably, using low doses of phagocytic target a recent study failed to detect a significant increase in phagocytosis by R522 variant expressing primary cells, but also suggested with a similar trend towards increased uptake (Takalo et al, 2020). Taken together, this could suggest that sustained and greater challenge of the PLCy2 signalling system leads to an impaired system, which is relevant given the in vivo context where continued stimulation of the pathway is expected and there appears to be a depletion of PIP2 in vivo. However, more work is required to understand this.

10 In figure 7 authors state that the experiments were done in 3 independent replicates. However, in panel C in some of the bars the SD are missing.

Thank you for bringing this to our attention, we have corrected the graph accordingly.

Reviewer 2

1. The authors describe that PLCG2 acts downstream of TREM2, a master regulator of microglial function, but do not provide evidence whether this is also true for the functional changes of the R522 mutant. Given that some microglial functions are TREM2-independent (see for example Andreone et al. Nature Neuroscience 2020), it is important to test TREM2-dependency, for example using the calcium or phagocytosis assays.

For the reasons outlined by the reviewer (also reviewer 1), we have now included experiments investigating the impact of the PLCy2-R522 variant downstream of TREM2. New experiments in Figure 3 C, D, and E analyse the resulting Ca²⁺ response in PLCy2 variant expressing cells following TREM2 activation using phosphatidyl-serine containing liposomes. Andreone *et al.* demonstrated how addition of these types of liposomes primarily exerted their downstream effects via PLCy2. Our experiments demonstrated increased TREM2 signalling, shown by an increased cytoplasmic Ca²⁺ increase, by cells expressing the R522 variant of PLCy2.

The changes made to the manuscript are highlighted in reply to point 5 of reviewer 1.

2. Figures 4 and 5: Phagocytosis of bacteria is reduced with the mutant, while endocytosis of Abeta and dextran is enhanced. The authors need to provide a mechanistic explanation for this discrepancy. And can they be sure that one process is phagocytosis, while the other one is endocytosis? If this is the real reason for the discrepancy, they should demonstrate this, for example using inhibitors of one or the other pathway.

It is correct that we found reduced phagocytosis of bacteria within the R522 in comparison to enhanced endocytosis of Abeta and dextran. The main pathway utilized for uptake of both zymosan and *E. coli* is receptor-mediated phagocytosis, with well-characterised and distinct receptor profiles recognising the two targets. Of note, one of the main receptors for zymosan on macrophages is dectin-1 (PMID: 12163569), an ITAM-like containing signalling molecule that signals through PLCy2 (PMID: 19136564). Dextran and amyloid, however, are believed to be taken into cells via alternate forms of endocytosis. To illustrate this, we have included data with a variety of inhibitors with differential effects on the uptake of the various cargoes (new Supplementary figure 2, please also see below).

Cytochalasin D, which inhibits actin cytoskeleton remodelling, was able to inhibit all forms of uptake as expected. Phagocytosis in particular is known to require substantial actin remodelling. On the other hand, chlorpromazine, which inhibits clathrin-mediated endocytosis, was able to almost completely abolish dextran/amyloid uptake whilst having minimal effect on uptake of zymosan/*E. coli*. In addition, methyl- β -cyclodextrin, which inhibits clathrin-independent endocytosis, was unable to inhibit uptake of zymosan/*E. coli* whilst significantly affecting dextran/amyloid uptake.

This data confirms what would be expected based on the literature: that E. coli and Zymosan are mainly taken up via phagocytosis, and dextran/abeta primarily via other forms of endocytosis.

The text in the manuscript has changed thus:

Results (lines 456-471):

"Inhibition of actin cytoskeleton remodelling using Cytochalasin D effectively reduced uptake of all examined cargos by >90% For all these inhibitors the effect was similar on cells with both variants."

3. Fig. 2C: Describe this experiment in more detail. Are there different kinetics (rapid rise for P522 and slow rise for R522)? Or is the trace not representative? And why is the increase so much stronger compared to the data in panels A and B?

Thank you for bringing this to our attention. In order to discount any variability across individual cells, we have replaced this data and show the means from 3 experiments (multiple cells per experiment) with error bars in place of the previous single traces for individual human iPSC microglia (figure 1 is appropriately altered). After doing the kinetics can be seen to be similar between the two variants, as in the mouse cells. We are sorry for the confusion we caused by showing single traces.

4. Fig. 3D: The authors need to better explain/speculate – at least in the discussion – how the increased Ab-induced calcium signalling may mechanistically be coupled to the increased Ab uptake in the phagocytosis assay.

We propose that the increased PLC γ 2 activity seen in the R522 variant after exposure to stimuli including amyloid oligomers is also seen in PLC γ 2 role in endocytosis of amyloid oligomers. We have added detail of a speculative mechanism in the discussion.

Discussion (line 688-697)

In addition, in contrast to what we see with phagocytosis, knockdown of PLC γ 2 does not fully inhibit endocytosis of either dextran or amyloid (supplementary Fig. 3). However, the decrease in endocytosis is greater in the cells with the R522 variant compared to the P522 variant. Taken together, this suggests PLC γ 2 only has a partial role in the uptake of these smaller particles (amyloid oligomers and dextran) which enter the cells by several endocytic systems (including CIE & CDE). Despite these observations, it is possible that the hyper function of the R522 variant is still directly responsible for the increased endocytic uptake in these cells. This could occur via the observed increased Ca²⁺ response seen in cells with the R522 variant, as this is known to function upstream of several endocytic systems (Nunes & Demaurex, 2010).

5. Fig. 6B: Describe in more detail why there is a difference in PIP₂ levels depending on whether LPS and Ab42 were preincubated or not. And why is the same not seen in panel 6D for macrophages versus microglia?

We apologise for not fully explaining our rationale for these experiments. In the preincubation experiments we attempted to stimulate a longer/stronger stimulation compared to the single dose experiments. We believe that the PIP₂ levels are depleted by increased activation of the R522 variant after stimulation therefore we proposed a second dose of LPS/amyloid would result in a greater depletion of PIP₂. We have not detected any compensatory response in other PIP species in these cells that would prevent a depletion of PIP₂ and we have detected basal depletion of PIP₂ in vivo.

We believe the general differences between microglia and the macrophages are broadly consistent with the thesis, with some differences that could be a consequence of different signalling dynamics between the two cell types coupled with the quite varied time courses of responses seen with the different stimuli.

Minor points:

7. Figure 1C: provide a magnification of the structure on the right, where you indicate the position of amino acid 522. The magnification will also allow a better visualization of the structural changes in the center of the structure.

This has been done, thank you for the suggestion.

Reviewer 3

In general, the manuscript is difficult to read and to understand because simple structure is missing e.g.: What are the main questions? What experiments were performed in order to answer the questions, what are the main results and what are the conclusions?

We are sorry that you found our manuscript difficult to read and understand, thank you for bringing this to our attention. We have made some changes to the introduction and discussion (see italicized below) to re-emphasize for the reader the main questions. The changes in the introduction, summarises the main questions and complements the individual list of questions that are retained in the manuscript (lines 59-64).

Introduction (line 103-107)

In this manuscript, we use specific CRISPR-mediated engineered alterations of human iPSC (hiPSC) and mice to ask how in physiologically-relevant cells (microglia and macrophages) and at appropriate expression levels does the protective R522 variant of PLC γ 2 influence cell signalling and classic functional activities, such as endocytosis?

Discussion (line 715-721)

In summary we have shown a consistent increase in PLC γ 2 enzymatic activation in novel human and mouse models due to the AD protective R522 mutation. This is associated with depletion of PI(4,5)P $_2$ *in vitro* after exposure to physiologically relevant stimuli. We also observe reduced phagocytic and increased endocytic clearance of multiple cargoes. We demonstrated that PLC γ 2 and PI(4,5)P $_2$ has a varied role in the uptake of these different cargoes. *In vivo* we have shown a basal decrease in mice with the R522 mutation in PLC γ 2 compared to wild type.

Although the authors would like to make a link to Alzheimer's disease especially in the second part of their study when they treated the cells with A β oligomers, this part of the study would definitely improve with *in vivo* experiments by crossing the PLC γ 2R522 mice with an AD mouse model.

We agree that crossing the PLC γ 2 R522 mice with an Alzheimer's disease mouse model would be extremely interesting and ongoing work aims to generate and characterize this model and address just these questions.

1) In the present study mice were housed in an environmental enrichment (EE), however it has been shown that EE can have an impact on microglia (Xu H et al. 2016 J Neurosci; Xu H et al. 2018 EMBO Molecular Medicine). How does housing the mice in an enriched environment affect the results?

Whilst there is evidence that deprivation of environmental enrichment (EE) can impact microglia, we routinely maintain environmental enrichment for two reasons: i) It is hoped that EE represents an ideal care scenario for most patients with Alzheimer's disease and hence most resembles the context in which we are attempting to decipher Alzheimer's pathology; ii) It is the default ethical stance for all research in the UK that animals be provided with EE wherever possible. Given this context, we do not see a strong argument for deprivation of EE as a routine point of study in this specific case.

2) In general, the quality of the Figures is lousy/poor and consequently the authors over-interpret their data. e.g. Figure 5 measurement of dextran in iPSC microglia is missing and needs to be shown, in Figure 6 there are no images for immunofluorescence shown only quantification is provided; images in supplementary Figure 2 are not convincing at all and images of A β are completely missing as well as images of microglia (only macrophages are shown). In Figure 8 only the intensity was measured for PIP2 levels and a real quantification is missing and needs to be done. From the images, it is questionable, if the PIP2 signal is really specific since in Supplementary Figure 2 it looks completely different.....Also, only images from the cortex are displayed but intensity was also measured in the hippocampus however representative images are absent here. Finally, morphological changes need to be detected in high resolution images in 3D and reconstructed with the image analysis software IMARIS.

We have added the measurement of dextran uptake in hiPSC as figure 5F. We apologise for the lack of representative images. We initially showed only some examples, but now shown, as supplementary Figure 6, we have included more images with different stimuli (including A β) of both macrophages and microglia. We have also added images from the hippocampus to figure 8 and additional images to supplementary figure 7. Quantification of PIP2 from in specific cells *in vivo* is somewhat challenging. We are unable to process brain lysates due the quantities of PIPs in other cells of the tissue and due to the likely alteration of PIP levels as an artefact of cell purification should we attempt to isolate microglia, hence we are somewhat constrained to *in situ* staining as a method to quantify PIP2.

We agree that there are differences in the appearance of the PIP₂ immunostaining *in vitro* and *in vivo*. Unfortunately, when immunostaining lipids differences in the fixation and staining methods can alter the staining patterns. For PIP₂ immunostaining this was demonstrated by Hammond et al 2009 PMID: 19508231. Similarly differing preparation protocols have been shown to alter PIP localisation (Omar-Hmeadi et al 2018 PMID: 29542271, Rajala et al 2014 PMID: 24964953, Hammond et al 2006 PMID: 16687737 and Hirono et al 2004 PMID: 15473969). Due to this we have not speculated on the localisation of PIP2 or on specific PIP2 pools in this paper.

Additionally, the possibility of morphological changes is of great interest and we are keen to conduct these experiments. Takalo et al. (PMID: 32917267) recently reported apparently normal microglial morphology in the Plcg2-R522 mice, but the numbers of mice studied were small and the experiment would be lacking in power to detect subtle alterations. It is our intent to conduct these studies with 3d analysis of microglia from a larger number of mice and also examine microglial behaviour under 2-photon imaging through cranial windows in awake mice.

3) Figure 4: iPSC microglia behave differently in C (*E. coli*) compared to F (Zymosan). There was already a significant difference after 1 hour detectable in F whereas in C only at the later time points 3+4 hours. Please explain.

As discussed above in response to the other reviewers, the main pathway utilized for uptake of both zymosan and *E. coli* is receptor-mediated phagocytosis, with well-characterised and distinct receptor profiles recognising the two targets. Of note, and as an example of the recognition receptor

differences between the two particles, one of the main receptors for zymosan on macrophages is dectin-1 (PMID: 12163569), an ITAM-like containing signalling molecule that signals through PLC γ 2 (PMID: 19136564). For this reason, we would expect a potentially more marked dependence on PLC γ 2 activity for the phagocytic uptake of zymosan and any perturbation of normal signalling would be expected to have a greater effect. We have added some text to the discussion noting this:

Discussion (line 633-636)

Zymosan and E.coli are phagocytosed by distinct receptor-mediated mechanisms, with a notable role for dectin-1 (Brown et al, 2002), an ITAM-like containing signalling molecule that signals through PLC γ 2 (Xu et al, 2009), in the clearance of zymosan.

4) Figure 5: In C only a graph is shown and no curve like in A and B. Why is this? Endocytosis in iPSC microglia should be measured over time the same way than in A and B.

Thank you for bringing this to our attention. Figure 5 has been updated so that uptake is measured over time in the iPSC-microglia instead of just at 2 hours.

5) Figure 8: the signal for PIP₂ looks completely different than the PIP₂ staining in the cells (Supplementary Figure 2). Why is this?

We believe this is due to differences in the fixation and preparation methods between the in vitro and in vivo samples. Please see our answer to this point in question 2 for more details.

6) Supplementary Figure 3 and Supplementary Figure 4 are not discussed in the results section at all.

With respect to Supplementary figure 4 we want to thank the reviewer for identifying this oversight which has been corrected. Supplementary figure 3 is referenced in the supplementary methods section as it is based on the protocols used here rather than the results generated. Due to other changes to the manuscript these figures have been renamed Supplementary figure 7 (previously Supplementary figure 4) and Supplementary figure 8 (previously Supplementary figure 3).

Results (line 518-522)

The cortex and hippocampus of two month old Plcg2^{R522} and Plcg2^{P522} mice were examined for PIP₂ levels in microglia by immunofluorescent staining for PIP₂ and Iba1 (Fig. 8, Supplementary Figure 7).

7) The discussion is too unfocussed. It is not clear what the main findings are and which ones are worth to discuss.

We are sorry that the reviewer found the discussion unfocussed we have made an effort to address this point in the paper (altered text is highlighted). We believe the main findings to be an increased enzymatic activation in novel human and mouse models due to the AD protective R522 mutation in PLC γ 2. This is associated with depletion of PI(4,5)P₂ both in vitro and in vivo and reduced phagocytic and increased endocytic clearance.

Dear Phil,

Thanks for submitting your revised manuscript to The EMBO Journal. Your study has now been seen by the referees and their comments are provided below.

As you can see while referee #2 is happy with the revised version, referee #3 still has some remaining concerns. I have discussed the remaining concerns further with referee #1.

Regarding the first point to cross PLcg2R522 mice with an AD mouse model. Both referee #1 and I find that while such an experiment would be insightful that it is not needed for the present study. The study is complete enough as is.

Regarding the 2nd point => quality of the figures. I find the overall quality of the figures good enough. However, the images shown in Figure 8a and supplementary figure 7 could be improved. Do you have more samples you can show or can you improve the quality of the images. Also would be nice to show several representative images - you can use the appendix figures for this. PS spelling mistake in figure 8A legend cyrosections.

We don't need further high-resolution images in 3D

When you resubmit will you also take care of the following points

We need a Data Availability section. This is the place to enter accession numbers etc. As far as I can see no data is generated that needs to be deposited in a database. If this is so then please state: This study includes no data deposited in external repositories. This section should be placed after the Materials and methods and before Acknowledgements

"Competing Interests" should be labeled as "Conflict of interest"

Please double check the reference list: more than 10 authors are listed. There is also a reference that starts with "P.Y " and called out in the text as P. et al. Just double check that this is correct.

There is a figure callout to Fig 2D, but no figure 2D. Callouts out to Fig 5F, Supp Fig 5 and Table 2 are missing

The appendix file is missing a ToC is missing. Please also correct figure naming and callout => it should be 'Appendix Figure S#' and 'Appendix Table S#'

You have a M&M section also in the appendix. I would prefer to have most of it in the main article. OK to have some M&M in the appendix in case you are providing very detail information, but the key parts should be in the main MS text.

Please check the ordering of the manuscript sections

We encourage the publication of source data, particularly for electrophoretic gels and blots, with the aim of making primary data more accessible and transparent to the reader. It would be great if you could provide me with a PDF file per figure that contains the original, uncropped and unprocessed scans of all or key gels used in the figure? The PDF files should be labeled with the appropriate

figure/panel number and should have molecular weight markers; further annotation could be useful but is not essential. The PDF files will be published online with the article as supplementary "Source Data" files.

We include a synopsis of the paper that is visible on the html file (see <http://emboj.embopress.org/>). Can you provide me with a general summary statement and 3-5 bullet points that capture the key findings of the paper?

I also need a summary figure for the synopsis. The size should be 550 wide by 400 high (pixels).

I have asked our publisher to do their pre-publication checks. I will forward you the comments as soon as I receive them.

That should be all. Let me know if we need to discuss anything further

With best wishes

Karin

Karin Dumstrei, PhD
Senior Editor
The EMBO Journal

When assembling figures, please refer to our figure preparation guideline in order to ensure proper formatting and readability in print as well as on screen:
<https://bit.ly/EMBOPressFigurePreparationGuideline>

Please remember: Digital image enhancement is acceptable practice, as long as it accurately represents the original data and conforms to community standards. If a figure has been subjected to significant electronic manipulation, this must be noted in the figure legend or in the 'Materials and Methods' section. The editors reserve the right to request original versions of figures and the

original images that were used to assemble the figure.

The revision must be submitted online within 90 days; please click on the link below to submit the revision online before 25th Jul 2021.

Referee #2:

The authors have adequately addressed my previous points.

Referee #3:

The revised manuscript „The R522 Alzheimer's protective PLCy2 hypermorph depletes substrate and alters cell function" was examined again. Although the authors have improved that manuscript and have provided additional information and data, some concerns remain:

1) Although the authors would like to make a link to Alzheimer's disease especially in the second part of their study when they treated the cells with A β oligomers, this part of the study would definitely improve with in vivo experiments by crossing the PLcg2R522 mice with an AD mouse model.

The authors do not provide new data by crossing PLcg2R522 mice with an AD mouse model as suggested. Therefore, I think the link to AD is not appropriate and far-fetched.

2) In general, the quality of the Figures is lousy/poor and consequently the authors over-interpret their data.

Although the authors provide some new images, these are not representative and still of poor quality. In addition, morphological changes detected in high resolution images in 3D and reconstructed with the image analysis software IMARIS as suggested are necessary and have not been performed.

Dear Philip,

Thank you for submitting your revised manuscript to the EMBO Journal. I have now had a chance to take a close look at everything and all looks good. I am therefore very pleased to accept the manuscript for publication here.

Congratulations on a nice study!

Best Karin

Karin Dumstrei, PhD
Senior Editor
The EMBO Journal

Please note that it is EMBO Journal policy for the transcript of the editorial process (containing referee reports and your response letter) to be published as an online supplement to each paper. If you do NOT want this, you will need to inform the Editorial Office via email immediately. More information is available here: https://emboj.embopress.org/about#Transparent_Process

Your manuscript will be processed for publication in the journal by EMBO Press. Manuscripts in the PDF and electronic editions of The EMBO Journal will be copy edited, and you will be provided with page proofs prior to publication. Please note that supplementary information is not included in the proofs.

Should you be planning a Press Release on your article, please get in contact with embojournal@wiley.com as early as possible, in order to coordinate publication and release dates.

If you have any questions, please do not hesitate to call or email the Editorial Office. Thank you for your contribution to The EMBO Journal.

Corresponding Author Name: Phil Taylor

Journal Submitted to: EMBO J

Manuscript Number: EMBOJ-2020-105603